# Inflammatory Intracellular Signaling in Neurons Is Influenced by Glial Soluble Factors in iPSC-Based Cell Model of *PARK2*-Associated Parkinson’s Disease

**DOI:** 10.3390/ijms25179621

**Published:** 2024-09-05

**Authors:** Tatiana Gerasimova, Daniil Poberezhniy, Valentina Nenasheva, Ekaterina Stepanenko, Elena Arsenyeva, Lyudmila Novosadova, Igor Grivennikov, Sergey Illarioshkin, Maria Lagarkova, Vyacheslav Tarantul, Ekaterina Novosadova

**Affiliations:** 1Laboratory of Translative Biomedicine, Lopukhin Federal Research and Clinical Center of Physical–Chemical Medicine of Federal Medical Biological Agency, 119435 Moscow, Russia; lagar@rcpcm.org; 2Laboratory of Molecular Neurogenetics and Innate Immunity, National Research Centre “Kurchatov Institute”, 123182 Moscow, Russia; d.poberezhniy1@gmail.com (D.P.); katishsha@mail.ru (E.S.); arslena@mail.ru (E.A.); novosadova-l@rambler.ru (L.N.); grivigan@mail.ru (I.G.); ninaslava130@yandex.ru (V.T.); novek-img@mail.ru (E.N.); 3Research Center of Neurology, 125367 Moscow, Russia; snillario@gmail.com

**Keywords:** Parkinson’s disease, *PARK2* gene, induced pluripotent stem cells, neurons, glia, neuroinflammation

## Abstract

Neuroinflammation is considered to be one of the driving factors in Parkinson’s disease (PD). This study was conducted using neuronal and glial cell cultures differentiated from induced pluripotent stem cells (iPSC) of healthy donors (HD) and PD patients with different *PARK2* mutations (PD). Based on the results of RNA sequencing, qPCR and ELISA, we revealed transcriptional and post-transcriptional changes in HD and PD neurons cultivated in HD and PD glial-conditioned medium. We demonstrated that if one or both of the components of the system, neurons or glia, is Parkin-deficient, the interaction resulted in the down-regulation of a number of key genes related to inflammatory intracellular pathways and negative regulation of apoptosis in neurons, which might be neuroprotective. In PD neurons, the stress-induced up-regulation of *APLNR* was significantly stronger compared to HD neurons and was diminished by glial soluble factors, both HD and PD. PD neurons in PD glial conditioned medium increased *APLN* expression and also up-regulated apelin synthesis and release into intracellular fluid, which represented another compensatory action. Overall, the reported results indicate that neuronal self-defense mechanisms contribute to cell survival, which might be characteristic of PD patients with Parkin-deficiency.

## 1. Introduction

Parkinson’s disease (PD) is one of the most widespread neurodegenerative diseases. It is characterized by the death of dopaminergic (DA) neurons in the substantia nigra manifested by clinically severe neurological disorders. About 10% of PD cases are inherited, and mutations in more than 20 genes have been identified that cause monogenic forms of PD. *PARK2* mutations are strictly linked to the familial form of PD [1]. The origin of most sporadic PD cases is unknown.

The main features of both sporadic- and genetics-related cases of PD are misfolded α-synuclein accumulation, mitochondrial dysfunction, oxidative stress, and neuroinflammation [2,3,4]. Determination of the role of Parkin in the above key processes is important for understanding the molecular basis of PD. Nevertheless, PD pathogenesis is considered to be not a pathology of only DA neurons, but also to include disrupted interactions between different cell types in the brain tissue [5].

Astrocytes are the most abandoned cell type in the central nervous system (CNS). These cells are proven to have a multifaced influence on the survival and function of neurons in healthy brains and in neurodegenerative disorders. First of all, they physically interact with neurons through compact morphological structures as a part of neuropil, which is composed of dendrites, axons, synapses, glial cell processes, and microvasculature. This is the mechanical support of nervous tissue, which is performed by astrocytes anchored to blood vessels, neurons, and other cells [6]. The second mechanism is related to secretory activity of astrocytes and humoral influence to other cell types [7,8]. The release of secreted molecules occurs by three main pathways: diffusion through transmembrane pores, transportation by transporters, and exocytosis.

The discovery of numerous signaling molecules synthesized by astrocytes led to the introduction of the special term—gliocrine system [9]. The meta-analysis performed by Jha et al. revealed 1169 proteins with high confidence, mainly in the secretome of primary astrocytes [10]. These molecules protect neurons against apoptosis, oxidative stress, and toxicity [7], promote functionally active CNS synapses, regulate the permeability of blood vessels [6], act as metabolic and trophic factors [9], mediate water and ion homeostasis [6,10], regulate adult neurogenesis [6,11,12], and also take part in inflammatory response [7,13]. Taken together, in neurodegenerative diseases, reactive astrocytes contribute to neuroprotection and regeneration on the one hand, and support detrimental long-lasting neuroinflammation on the other hand [5,10]. Inflammatory cytokines actively produced by microglia and astrocytes initiate pro-inflammatory signaling, which results in the neurotoxicity associated with PD [1]. In a healthy brain, the activity of pro- and anti-inflammatory cytokines is a finely balanced system which is dysregulated in PD [14].

Many signals can drive glial reactivity, as follows: molecules of injured neurons, microglia-derived pro-inflammatory cytokines, and pathological proteins. Such a wide variety of stimuli, combined with high phenotype plasticity of astrocytes, drives the cells to many distinct activation states lying on a continuum between pro-inflammatory A1 and anti-inflammatory A2 types. It has been highlighted that many populations of astrocytes with diverse phenotypes exist [15]. Parkin deficiency has recently been connected to “mito-inflammation”, inflammation that arises through mitochondrial pathways [1]. Since it has been shown that *PARK2*-associated PD is less dependent on the level of synuclein accumulation than PD with mutations in other genes [16], the role of astrocyte-derived cytokines and chemokines might have greater significance in this case.

Interestingly, astrocytes can become reactive even after being cultivated in monoculture in vitro that is in the absence of any incoming stimuli. One of the possible explanations for this is the ability of astrocytes to respond to self-secreted signaling molecules by binding to receptors on the astrocyte cell membrane. Another fact is that in the brain tissue, neighboring cells produce so-called “resting signals” keeping glia in a steady state. These signals are absent in a culture dish that cause slight signs of reactivity in astrocytes [17]. This fact perhaps makes the cell culture model closer to an in vivo state, since different low-level stimuli are continuously acting in vivo, which determines the basal level of cell activation.

Generation of specialized cells from induced pluripotent stem cells (iPSC) represents a promising tool for studying patient-specific human neurons and glia, since it is almost impossible to obtain live cells from human brain tissue. The method allows for obtaining genetically unique cells in quantities sufficient for research, which is an undeniable advantage. iPSC-derived cell cultures are useful for modeling neurodegenerative disorders, for a deeper understanding of pathogenetic mechanisms, and also for the identification of novel therapeutic targets [18]. In addition, the iPSC-based approach is useful for studying the influence of PD-associated mutations on cell functions, both in mixed cultures and in any cell type separately.

The method of co-cultivating neurons and glia has advantages, modeling the presence of different cell types in the system. However, it confounds the direct contacts between the cells and the influence of soluble factors [19]. We planned the experiment in accordance with the goal of evaluating the impact of astrocyte-derived proteins on the neurons’ gene network and of identifying the differences between HD and *PARK2*-mutant neurons in their interactions with glial soluble factors of HD and PD patients. We also made an attempt to determine which of these features could be directly related to the genotype of neurons and which could be influenced by the secretome of astrocytes.

## 2. Results and Discussion

### 2.1. Characterization of the Initial Cell Composition and Differentiation Status of the Cell Cultures Used in This Study

#### 2.1.1. Neuronal Cell Cultures

Induced pluripotent stem cells (iPSCs) obtained from a healthy donor (HD) and from a PD patient with *hom EX8 del PARK2* (PD) underwent the full course of differentiation into mature neurons, as described previously [20]. The characteristics of all the cell lines used in this study are given in Table 1.

The qPCR data show that both HD and PD neuronal cultures expressed markers of mature neurons on similar levels (*NEUN*, *MAP2*, *TUBB3*, and *AADC*) and therefore were suitable for comparison (Figure 1A). In addition, expression of the TUBB3 protein was estimated using immunocytochemistry, and the marker was displayed by 93–95% of cells (Figure 1C).

#### 2.1.2. Glial Cell Cultures

iPSCs obtained from two healthy donors and two patients with *PARK2*-associated PD (*het EX2 dup PARK2*; *het EX2 del PARK2*) (Table 1) underwent the full course of differentiation in glial direction according to the protocol [24].

The resulting cells expressed mature astrocytes’ markers (*ALDH1L1*, *GLUT1*, and *VIM*) with no significant difference between HD and PD cells based on the qPCR results (Figure 1B). Glial cultures were also stained with antibodies to astrocyte-specific marker S100 and were shown to contain 94–97% S100-positive cells (Figure 1D).

Since the secretory profile of individual astrocyte cell lines could be expectedly individual, we prepared mixed conditioned medium obtained from two HD glial cultures and mixed conditioned medium obtained from two PD glial cultures with *PARK2* mutations (see Section 3). Furthermore, we cultivated HD neurons and *PARK2*-defitient PD neurons in those glial mixed media in four possible combinations (Figure 2). As a control, we used the HD and PD neurons in the base neuronal medium and HD and PD neurons in the base glial medium. After 72 h of cultivation, the cells were collected for subsequent RNA isolation and RNA sequencing. For ease of understanding, the following alphanumeric notations will be used further through the article:○HD0—HD neurons in the base neuronal medium.○PD0—PD neurons in the base neuronal medium.○HD1—HD neurons in the base glial medium.○PD1—PD neurons in the base glial medium.○HD2—HD neurons in the HD glial mixed medium. ○PD2—PD neurons in the HD glial mixed medium. ○HD3—HD neurons in the PD glial mixed medium.○PD3—PD neurons in the PD glial mixed medium.

### 2.2. Full-Transcriptome Profiling Using High-Throughput RNA Sequencing (RNAseq) and Bioinformatics Data Analysis

Visualization with Principal Component Analysis (PCA) showed that HD neurons in different types of media were clustered next to each other but separately from PD neurons (Figure 3).

The simple pairwise comparisons of the obtained RNAseq data revealed a number of differentially expressed genes (DEGs). Surprisingly, comparisons involving HD neurons demonstrated significantly higher numbers of DEGs compared to those with PD neurons (Table 2).

Within the framework of the experiment, differential gene expression can be explained by the influence of two factors: Medium (M) and Disease Status (DS) of neurons (DS). In this regard, a model described using the formula ~ M + DS + M × DS was chosen, which allowed us to take into account the influence of both factors and of their interaction (M × DS) as well.

We used an appropriate “baseline” level of gene expression according to the two types of comparison. To compare the gene network in neurons cultivated in different types of media within the Disease Status groups (HD or PD), gene expression in neurons in a base glial medium (HD1 and PD1) was used as a “baseline”. For example, the comparison (HD3vsHD1) vs. (HD2vsHD1) reflects the differences in gene expression between HD3 compared to HD1 and HD2 compared to HD1. The comparison (PD0vsHD0) vs. (PD1vsHD1) points to differences in gene expression between PD and HD neurons in base neuronal medium and PD and HD neurons in base glial medium (Table 3).

The Venn diagrams below demonstrate that HD glial factors have a more pronounced effect on the neuronal gene network compared to PD secretome, which correlates with the pairwise comparison results (Figure 4).

Gene Set Enrichment Analysis (GSEA) and Over Representation Analysis (ORA) were conducted, and the lists of the enriched pathways associated with inflammation were obtained for all comparisons (Table 4 and Appendix A). It is important to note that, according to the results of both GSEA and ORA, in PD0vsHD0, not one of the mentioned inflammatory pathways was enriched, which allowed us to consider the differences in other comparisons not dependent on the individual genotypes of neuronal cell lines, but related to the joint influence of neurons’ diseased status and glial medium type.

Differences were observed in comparisons between the HD system (HD neurons in HD glial mixed medium) and those systems in which one of the components or both glia and neurons were Parkin-deficient. The richest composition of inflammatory pathways was displayed in PD2vsHD2, i.e., PD and HD neurons in HD glial mixed medium. On the contrary, comparisons between PD3vsHD3 and PD3vsPD2 did not show any significant differences in inflammatory gene network. One could conclude that in the field of inflammation, the HD cell system was more reactive than the PD one, as neurons and glia obtained from the iPSCs of healthy donors had greater ability to interact with each other.

Surprisingly, the majority of DEGs related to inflammatory pathways were down-regulated in all of the comparisons: the ratio of up- and down-regulated genes was 17/72 in PD3vsHD2, 18/70 in PD2vsHD2, and 16/49 in HD3vsHD2. Among the down-regulated DEGs, important pro-inflammatory (*JUN*/*FOS*/*JUNB*, *F2RL1*, *ICAM1*/*VCAM*, *CXCL12*/*CXCR4*, *ATF3*/*DDIT*, etc.), anti-inflammatory (*TNFAIP3*, *SOCS3*, *PER1*, *DUSP1*, *ZFP36*, etc.), and also regulatory (*ATF3*, *BCL3*, etc.) genes were found. The results indicate a general decrease in the activity of genes involved in inflammatory signaling when PD neurons interact with HD glial mixed medium and HD neurons with PD glial mixed medium.

The main intracellular inflammatory signaling is canonical NF-kB, the activation of which leads to a massive synthesis of pro-inflammatory cytokines and chemokines. The pathway was not enriched by GSEA and ORA, and there were no DEGs encoding pro-inflammatory cytokines in any comparison. With the aim to consider the pathway more carefully, we assumed that some genes might not pass the cutoff of FC = 1.5 but, nevertheless, change their expression. To track the dynamics of the pathway as a whole, we put the complete list of genes on KEGG (Kyoto Encyclopedia of Genes and Genomes) graphs, with the DEGs given in color (Appendix A). Differences in NF-kB genes’ expressions were observed between PD and HD neurons in HD glial mixed medium, but not in PD glial mixed medium. HD neurons, but not PD neurons, reacted to glial medium variants. The differences mostly were represented by a decreased expression in a number of NF-kB genes, which is consistent with the general observations described above.

### 2.3. Identification of DEGs Related to the Disease Status of Neurons or to the Glial Medium Type

We compared DEGs from all of the enriched inflammatory pathways in PD3vsHD2 with those in PD2vsHD2 and revealed the genes that were differentially expressed in both comparisons, which meant that they did not depend on the glial medium and were associated with the disease status of neurons. These genes were subtracted from DEGs lists in every other comparison. We considered that the genes remaining in PD2vsHD2 reflect the influence of HD glial mixed medium on PD neurons, and the genes remaining in HD3vsHD2 reflect the effect of PD glial mixed medium on HD neurons (Table 5). Almost all of the genes were down-regulated with FC in the range from 2 to 16, and also appeared to be linked by the literature data to neurodegenerative diseases, PD pathogenesis, and/or Parkin.

**Group 1:** DEGs associated with the disease status of neurons.

This group had the richest composition of hub genes, mostly participating in several signaling pathways simultaneously (Table 5). No clear directionality of the DEGs list, inflammatory or anti-inflammatory, was observed. Expression of several pro-inflammatory genes (*CXCL12*, *CXCR4*), negative regulators of inflammation (*TNFAIP3*, *SOCS3*, *DUSP1*), and regulatory genes (*BCL3*) that mediate their activity through NF-kB and MAPK signaling, were suppressed. Notably, MAPK and CXCL12/CXCR4, but not NF-kB, signaling pathways were enriched in both comparisons PD3vsHD2 and PD2vsHD2.

Dysregulated NF-kB, MAPK signaling and CXCL12/CXCR4 axis were previously shown to contribute to PD and other neurodegenerative diseases [26,38,52]. According to accumulated evidence, expression levels of pro-inflammatory *CXCL12*, *CXCR4*, *ATF3*, and *BCL3* in PD were found up-regulated, while levels of anti-inflammatory *TNFAIP3* and *SOCS3* decreased. Other genes (*ICAM1*, *HLA-B*, and *HLA-C*) were shown to be strongly activated in the presence of α-synuclein and TNF-α [15,72,73]. All of the genes from Group 1 were down-regulated in PD neurons.

Evidence on the interrelation of some of the Group 1 genes with Parkin exists (Table 5). NLRP3 inflammasome promoted by DDIT3 [27] is a substrate for ubiquitination by Parkin [28]. Parkin overexpression induces Gadd45a level [48]. In the absence of Parkin, high levels of mitochondrial antigens are presented on MHC class I molecules in macrophages and dendritic cells [32], and, at the same time, neurons can also express MHC-I molecules following exposure to inflammatory cytokines [73]. 

**Group 2:** DEGs characteristic for PD neurons in HD glial mixed medium.

Several positive regulators of cytokine gene expression, particularly members of transcription factor activator protein 1 (AP-1), were found to be down-regulated among the DEGs in this group (*FOS*, *JUNB*). According to the literature data, the induction of AP-1 complex and *CEBPB* appears to be essential for the expression of several cytokines such as IL-1, IL-2, IL-6, TNF-α, IFN-γ, IL-8, and MIP-α [41]. At the same time, in Group 2, *TGFB1*, known to decrease neuroinflammation in PD [37], and *PTGDS*, an anti-inflammatory arm of arachidonic acid cascade [62,63], were also down-regulated.

**Group 3:** DEGs characteristic for HD neurons in PD glial mixed medium.

One of the few up-regulated DEGs in this group was *A2M*. The gene encodes a protein that acts as a carrier for different molecules, and in particular binds pro-inflammatory cytokines (TNF-α and IL-1α), thus playing an important role in cytokine clearance [70]. *A2M* was significantly overexpressed, which might be a response to the higher concentration of pro-inflammatory cytokines in PD glial mixed medium in comparison to HD glial mixed medium.

Several DEGs in the group were shown to interact with Parkin. One of them is *MAOB*. The published data suggest that *MAOB* induction can interfere with mitochondrial quality control via loss of Parkin activity [74]. Otherwise, Parkin may limit the expression of MAO and thus restrict the ROS produced during dopamine oxidation by MAO [69].

Another DEG related to Parkin, *HSPA5*, was found in both Group 2 and Group 3. The gene’s product, heat shock 70 kDa protein 5, is needed to maintain the quality control of proteins in ER and suppress α-synuclein aggregation and toxicity in PD models [61]. The STRING tool (v 12.0) provides evidence of the protein–protein interactions between HSPA5 and Parkin detected using an affinity chromatography technology assay with an average detection confidence of ‘exploratory’. The same STRING data are presented regarding the TRFC protein’s connection with Parkin. Thus, our results support these observations.

Interestingly, among the DEGs in Group 2 and Group 3, unlike Group 1 associated with neuron status, we found receptors to growth factors (*FGFR3*, *IGF2*, and *TGFB1*) that presented an upstream control of the MAPK signaling pathway [60]. It is important to note that the influence of base glial medium, containing FGF2 and IGF1, was subtracted in the bioinformatic analysis.

Taken together, if one or both of the components of the system, neurons or glia, lacks in Parkin, their humoral interaction leads to the general suppression of the inflammatory gene network in neurons compared to the HD system.

The protein–protein interaction between the products of key DEGs and Parkin (PRKN) were visualized using the STRING tool for each group (Figure 5). A close connection between the proteins in inflammatory signaling is obvious.

In order to validate and support the RNAseq data, the expression of some DEGs from each group was estimated using qPCR (Appendix A). Linear regression analysis revealed a robust correlation between RNAseq and qPCR data on the expression of these DEGs (Figure 6).

Additionally, we repeated the experiment using one more PD neuronal culture obtained from iPSCs with another mutation in the *PARK2* gene (*het EX2 del PARK2*). The results of qPCR on the expression of the same genes supported our observations (Appendix A).

### 2.4. Determination of Cytokine/Chemokine Composition of Glial Mixed Supernatants

Astrocytes produce a wide range of signaling molecules, and the complete composition of supernatants cannot be determined. Considering that astrocytes were proven to be immunocompetent cells, we set to the task of investigating HD and PD glial mixed supernatants on the presence and concentrations of some main proteins associated with inflammation.

According to the results of a multiplex protein analysis, the concentration of multiple cytokines and chemokines with pro-inflammatory activity and of two anti-inflammatory cytokines were increased in the mixed supernatant obtained from PD astrocytes in comparison to the HD glial mixed supernatant. Some of the cytokines (IL-1b, IL-10) presented in the PD mixed supernatant were below the detection level in the HD one (Figure 7).

In order to make sure that not only the PD glial mixed supernatant, but both individual PD glial supernatants contained higher levels of cytokines than both HD ones, we conducted an additional round of multiplex protein analyses with supernatants from individual glial cell cultures (two HD and two PD) taken in two biological replicates (Appendix A). The results correlate with the concentration ratio between the HD and PD mixed supernatants.

Next, we explored the neuronal expression of the receptors to the chosen cytokines and chemokines, since, for intracellular signaling initiation, two components, ligands and cell surface receptors, are needed.

### 2.5. Determination of the Neuronal Expression of Genes Encoding Receptors to Corresponding Glial-Derived Cytokines Found in HD and PD Glial Mixed Supernatants

Based on RNAseq data analysis validated by qPCR, we observed that (i) there were no initial differences between HD and PD neurons in the expression of the genes encoding receptors to the studied glia-derived cytokines; (ii) the expression of those genes was also not affected by the type of culture medium, and thus no signs of feedback regulation were detected.

The only gene from those encoding cytokine receptors that demonstrated slight but statistically significant changes between comparison groups was *TNFRSF1B* (Figure 8). TNF-α functions via two different receptors, TNFR1 (encoded by gene *TNFRSF1A*) and TNFR2 (*TNFRSF1B*). Neurons constitutively produce both receptors. TNFR1 activates inflammatory signaling, and TNFR2 up-regulates pro-survival signals and reinforces TNFR1-mediated inflammation [75]. TNFR2 is associated with the activation of non-canonical NF-κB [14]. We found that PD glial mixed medium significantly up-regulated *TNFRSF1B* in HD neurons, whereas the gene level was similar in HD and PD neurons cultivated in HD glial mixed medium (Figure 8).

### 2.6. Apelin/APLNR System

Our attention was drawn to differences in the expression of genes associated with apelin/APLNR signaling. Based on the literature data, the G-protein-coupled receptor APLNR is present in the neurons and glial cells of different brain areas, including the substantia nigra [3,76,77,78]. Apelin-13 is the most active isoform among APLNR ligands. The apelin/APLNR axis acts as a stress response in various diseases and plays a modulatory role in neurodegenerative disorders as it normalizes apoptosis, autophagy, oxidative stress, and exhibits anti-inflammatory properties in the CNS [3,77,78,79,80,81,82]. The apelin/APLNR system has been the focus of increased attention in studies aimed at elucidating the mechanisms of PD and searching for therapeutic targets. Most of them are based on animal models or on neuroblastoma cell line SH-SY5Y [3,78]. In our research, we demonstrated the influence of glia-derived factors on the functioning of DA neurons using genetically unique human cell cultures of HDs and patients carrying different *PARK2*-mutations.

We investigated all of the components of our system on transcriptional and translational levels: *APLN* and *APLNR* gene expression in neurons, apelin protein level in glial mixed supernatants (glia-derived apelin), and apelin synthesized and released into the medium by neurons (neuron-derived apelin).

Based on the RNAseq and qPCR results, the initial levels of *APLNR* expression were similar in PD and HD neurons (HD0—PD0). *APLNR* transcription raised after the neuronal culture medium changed to base glial medium (HD1—PD1), which was probably a kind of stress for the cells. Notably, such a reaction in HD neurons was significantly weaker than that in PD cells. Soluble HD and PD glial-derived factors significantly down-regulated *APLNR* expression in PD neurons. Compared to HD neurons, PD cells demonstrated significantly higher *APLNR* expression when cultured in HD glial mixed medium and lower expression in PD glial mixed medium (Figure 9A).

*APLN* expression was up-regulated in PD3, while in HD neurons it remained stable in all of the medium types (Figure 9B).

Next, apelin synthesis in neurons was determined by ELISA. We compared the levels of apelin secreted by HD and PD neurons into intercellular fluid and the amounts of the protein deposited inside the cells.

The results demonstrated that the base level of the studying regulatory peptide in PD neurons in the neuronal medium was close to HD cells, which corresponded to qPCR data. Apelin intracellular content remained approximately the same both in HD and in PD neurons in all of the four medium types. The amount of apelin secreted by HD neurons demonstrated a wide variability between biological replicates, and overall was not dependent significantly on the medium type. On the contrary, PD neurons increased apelin synthesis many times when supplemented with control medium and significantly down-regulated it under the influence of glial soluble factors (Figure 10A,B).

The mechanism of β-arrestin-dependent APLNR desensitization and endocytosis is reported [76]. Sustained activation of APLNR by its ligands can cause desensitization [83], so the receptor expression depends on the apelin concentration in intercellular fluid. These data support our results, as the lowest level of *APLNR* expression was observed in PD3, which might be explained by the feedback principle, since the level of the apelin in PD glial mixed medium was significantly higher than in the HD glial mixed medium (Figure 9C).

The evidence linking the apelin system to inflammation is quite strong. Pro-inflammatory stimuli up-regulate apelin expression. For example, in ileal cell culture experiments, treatment with IL-6 or LPS led to increased apelin expression [81]. In cultured rotator cuff cells, apelin mRNA levels were significantly up-regulated after TNF-α stimulation [84]. TNF-α also induced the expression of the apelin receptor in HepG2 cells [85]. This corresponds to a higher level of apelin and pro-inflammatory cytokines in PD glial mixed supernatants compared to HD ones.

In the published data, increased production of pro-inflammatory cytokines in Parkin deficiency is commonly associated with impaired mitochondrial function and oxidative stress, which indicates an indirect Parkin influence. The direct role of Parkin in neuroinflammation is not so obvious and is poorly covered in the literature. The reports mostly concern microglia and astrocytes, but not neurons. Wang et al. proved that upon TNF-α stimulation, Parkin ubiquitinylates RIPK1, which promotes the activation of MAPKs and NF-κB with p65 nuclear translocation in astrocytes [86]. Recently, we reported that stimulation of glial cell lines by TNF-α caused a more pronounced inflammatory response in HD glia than in Parkin-deficient PD glia, showing less reactivity potential of *PARK2*-deficient cells, which was in accordance with the above Parkin action [23]. It is important to note that all of the abovementioned data were obtained using the experimental stimulation of astrocytes by inflammatory cytokines, mostly by TNF-α, that modeled M1 microglia’s influence in PD brain tissue.

The assumption was proposed that inflammatory cytokines (TNF-α and IL-1β) can have protective effects when conditions of duration, amount of expression, or state of activation of the target cells are changed [87]. In the present study, supernatants from resting glia were used so that the medium contained about 1000 times less TNF-α than in the case of stimulation experiments. In these conditions, the down-regulation of NF-kB signaling might present a neuronal self-protective mechanism. This hypothesis is supported by the literature data. A comprehensive review by Anilkumar et al. discusses in detail the diverse effect of NF-kB on different cell types within the CNS [88]. The authors point to the dual role of the signaling in neuronal survival, neuroprotective in several cases, but not in neurodegeneration. The TNF-α/NF-κB pathway is upstream of p53-dependent apoptosis. In a recent study, *p53 KO* improved the survival of engrafted iPSC-derived DA neurons [89]. In our study, NF-kB suppression was found in HD3 vs. HD2, in PD2 vs. HD2, and in PD3 vs. HD2, in case one of the cell types or both were Parkin-deficient. Moreover, the pathway associated with the negative regulation of apoptosis (GO:0043066) was enriched in all of the mentioned comparisons based on ORA results (Appendix A). Further research is needed to indicate if such self-protection is characteristic for PD in general or for Parkin deficiency only.

It is shown that the activation of APLNR signaling inhibits neuronal apoptosis by increasing the ratio of Bcl-2/Bax and decreasing cleaved caspase-3 expression [90], which has a pro-survival effect on neurons. Since activation of the apelin/APLNR system might have a protective reaction on stress, it could be assumed that this mechanism was more active in PD neurons than in HD cells, or that the change of culture medium was significantly more stressful for PD neurons than for HD neurons. Moreover, PD cells actively reacted to stress by strongly up-regulating *APLNR*. HD glial factors slightly down-regulated, while PD glial factors depressed this effect, as *APLNR* expression in PD3 was even lower than in HD3. It can be assumed that a decrease in *APLNR* expression, and therefore a weakened signaling through the receptor, reflects a decrease in the neuroprotective capabilities of PD neurons in the presence of PD glial factors (i.e., in natural combination in PD). In such conditions, PD neurons up-regulated *APLN* transcription and actively released apelin into intracellular fluid. That might point to one more compensatory mechanism activated in neurons. On the one hand, neuron-derived apelin could bind to astrocyte receptors and suppress the release of inflammatory cytokines [80], acting as a feed-back humoral response to glial soluble factors. On the other hand, it could also interact with receptors on the surface of neurons and suppress endogenous NF-kB, since apelin inhibits the pathway [91]. Importantly, HD neurons showed no significant difference in *APLNR*/*APLN* expression and released apelin at very low levels, which meant that the influence of HD and PD glial soluble factors on the activity of the apelin/APLNR system is a characteristic feature of Parkin-deficient neurons.

Short summary of the results:(i)PD glial mixed media contained comparatively higher concentrations of a number of pro-inflammatory and anti-inflammatory cytokines and chemokines than HD glial mixed medium.(ii)The most genes encoding receptors to the studied cytokines were expressed on the same level in PD and HD neurons, and that was not affected by soluble factors of glial mixed media. Thus, no signs of feedback regulation were detected.(iii)The inflammatory pathways enriched in PD2 vs. HD2, in PD3 vs. HD2, and in HD3 vs. HD2 demonstrated differences dependent both on the neurons’ genetic status and on the glial soluble factors. Among the main enriched pathways, MAPK signaling and those related to cytokines (cytokine–receptor interactions, responses to cytokine, and cytokine production) were detected.(iv)The ‘HD neuron/HD glia’ system might be characterized as more reactive than the PD one, as HD neurons and soluble factors of HD glia demonstrated greater ability to interact with each other.(v)If one or both of the components, neurons or glia, is Parkin-deficient, humoral interactions between the cell types led to a general suppression of the inflammatory gene network in neurons compared to the HD system.(vi)*APLNR* expression was up-regulated in HD and PD neurons as a stress response. In cases of PD neurons, it was stronger and was accompanied by increased synthesis and release of apelin into intracellular fluid. In PD neurons, but not in HD cells, *APLNR* expression and apelin secretion were down-regulated by glial soluble factors, both HD and PD. PD glial secretome to the greatest extent depressed *APLNR* in PD neurons, which was accompanied by a high level of *APLN* transcription and apelin release.

## 3. Materials and Methods

### 3.1. iPSC Differentiation in Neuronal and Glial Direction and Cultivation of Neurons

The differentiation of iPSCs (in glial and neuronal direction) was performed as described earlier [20,24]. The characteristics of all of the cell lines used in this study are given in Table 1.

Glial cultures were cultivated in the base glial medium: DMEM/F12 (Gibco, Billings, MT, USA), 1% NEAA (Hyclone, Logan, UT, USA), Glutamine 2 mm (PanEco, Moscow, RF, Russia), N2, B27, FGF2 8 ng/mL, Heregulin 10 ng/mL, IGF1 200 ng/mL, Activin A 10 ng/mL (Stem Cell Technologies, Vancouver, BC, Canada) *w*/*o* antibiotics. Neurons were cultivated in the base neuronal medium: DMEM/F12 (Gibco, Billings, MT, USA), serum replacement 2% (Gibco, Billings, MT, USA), non-essential aminoacids 1 mM (PanEco, Moscow, RF, Russia), L-glutamine 2 mM (ICN Biomedicals Inc., Costa Mesa, CA, USA), penicillin-streptomycin 50 μg/mL (PanEco, Moscow, RF, Russia), B27 supplement 1% (Life Technologies, Carlsbad, CA, USA), forskolin 5 μM (Stemgent, Cambridge, MA, USA), BDNF 20 ng/mL, GDNF 20 ng/mL, ascorbic acid 200 μM (all from Peprotech, Cranbury, NJ, USA). Glial supernatants were collected and mixed—two HD supernatants 1:1 and two PD supernatants 1:1. As a result, two types of glial mixed media (HD and PD) were prepared. They contained 70% of the abovementioned glial mixed supernatants + 30% of base neuronal medium. Neuronal cultures were seeded on a 96-well plate 40,000 cells/well overnight. The next day, 4 types of media (base neuronal, base glial, HD glial mixed, PD glial mixed, and all of the media *w*/*o* antibiotics) were added to neurons in the volume of 200 μL/well. After 48 h, the supernatants were collected for multiplex/ELISA analysis, and cells were counted for the recalculation of protein concentrations per 10^6^ cells. For the qPCR analysis, cells were lysed with TriReagent (MRC, Cincinnati, OH, USA). For the RNAseq, neuronal cultures were cultured in 35 mm Petri dishes in the same 4 types of media, and, after 72 h, were collected with the Trizol reagent (Invitrogen, Carlsbad, CA, USA).

### 3.2. Full-Transcriptome Profiling Using High-Throughput RNA Sequencing (RNAseq) Technology and Bioinformatics Data Analysis

Total RNA was isolated from neuronal cultures using the Trisol reagent and the PureLink RNA micro Kit (Invitrogen, Carlsbad, CA USA), following the manufacturer’s instructions. The quality and quantity of isolated total RNA was checked using a BioAnalyser and the RNA 6000 Nano Kit (Agilent, Santa Clara, CA, USA). Next, the polyA fraction was obtained using a oligoT magnetic beads Dynabeads^®^ mRNA Purification Kit (Ambion, Austin, TX, USA) according to the instructions for the kit. Libraries were then prepared from polyA RNA for massively parallel sequencing using the NEBNext^®^ Ultra™ II RNA Library Prep kit (NEB, Ipswich, MA, USA) according to the kit’s instructions. Library concentrations were determined using the Qubit dsDNA HS Assay Kit (Thermo Fisher Scientific, Waltham, MA, USA) on a Qbit 2.0 instrument. The library fragment length distribution was performed using the Agilent High Sensitivity DNA Kit (Agilent, Santa Clara, CA, USA). RNA sequencing of the samples was performed on the Illumina platform (San Diego, CA, USA) with a generation of at least 10^6^ short reads (50 nucleotides). Raw sequencing data were converted to FASTQ format using bcl2fastq software v 2.20 (Illumina). Quality control of the reads was carried out using the FastQC program (v 0.12.1) [92]. Adapter removal, quality trimming, and length filtration of reads was performed using Cutadapt (v 2.6) [93]. Mapping and quantifying of reads at the levels of individual transcripts and genes was implemented using Salmon (v 1.10.2) and the tximport (R package, v 1.30.0) [94,95] with Homo sapiens reference genome (GRCh38.14) and annotation (release 44). Further data processing was carried out using DESeq2 (R package, v 1.42.0) [96] (Appendix A). The functional analysis of the obtained DEGs lists was carried out by implementing Over-Representation Analysis (ORA) with g:Profiler Toolset (v e110_eg57_p18_4b54a898) via R interface gprofiler2 (v 0.2.2) [97], and Gene Set Enrichment Analysis (GSEA) was carried out with the fgsea R package (v 1.28.0) [98] using the Gene Ontology v 2023-11-15 (GO) [99] and Kyoto Encyclopedia of Genes and Genomes (KEGG) [100] databases. Protein–protein interaction networks’ construction and analysis were carried out using the STRING database [101].

### 3.3. Immunocytochemistry

Adherent cells on the Petri dish were washed with PBS, fixed with 4% paraformaldehyde in PBS (pH 6.8) for 20 min at room temperature (RT), and washed in PBS with 0.1% Tween 20 (Sigma-Aldrich, Saint Louis, MO, USA) three times for 5 min. Nonspecific antibody sorption was blocked by incubation in blocking buffer (PBS with 0.1% Triton ×100 and 5% fetal bovine serum (HyClone, Logan, UT, USA)) for 30 min at RT. Primary antibodies (Table 6) were applied overnight at 4 °C and then washed in PBS with 0.1% Tween 20 three times for 5 min. The secondary antibodies were applied for 60 min at RT, then washed in PBS with 0.1% Tween 20 three times for 5 min. After that, the cell cultures were incubated with 0.1 μg/mL DAPI (Sigma-Aldrich, Saint Louis, MO, USA) in PBS for 10 min for visualization of the cell nuclei and then were washed twice with PBS. The cells were investigated using the AxioImager Z1 fluorescence microscope (Carl Zeiss, Oberhohen, Germany), and images were taken with AxioVision 4.8 software (Carl Zeiss, Oberhohen, Germany). For cell counting, the multiple fields that cover the whole dish’s surface were imaged. The obtained images were analyzed with ImageJ 1.49 software (NCBI, Bethesda, MD, USA) using ITCN plugin (Center for Bio-image Informatics, Santa Barbara, CA, USA).

### 3.4. Multiplex Analysis of Proteins

Concentrations of cytokines and chemokines (EGF, Eotaxin, GM-CSF, G-CSF, IFNα2, IFNγ, IL-10, IL-12P40, IL-12P70, IL-13, IL-15, IL-17A, IL-1RA, IL-1α, IL-1β, IL-2, IL-3, IL-4, IL-5, IL-6, IL-7, IL-8, IP-10, MCP-1, MIP-1α, MIP-1β, RANTES, TNFα, TNFβ, VEGF, FGF-2, TGF-α, FLT-3L, Fractalkine, GRO1, MCP-3, MDC, PDGF-AA, PDGF-AB/BB, sCD40L, and IL-9) were measured in two glial mixed supernatants and in supernatants from individual glial cultures using a multiplex bead array with the 41-plex human cytokine/chemokine panel (HCYTMAG-60K-PX41, Merck, Darmstadt, Germany). Antibody immobilized beads were detected on the multiplex reader Bio-Plex “MAGPIX” (Bio-Rad, Hercules, CA, USA). The results were presented as mean ± SEM. A statistical analysis of the data was performed using an unpaired two-tailed *t*-test (GraphPad Prism 8.0.1. software). The differences were considered to be statistically significant at * *p* < 0.05, ** *p* < 0.01, *** *p* < 0.001, **** *p* < 0.0001.

### 3.5. ELISA

The concentration of apelin in culture media and in cell lysates (in RIPA containing inhibitors of proteases) was estimated using a human Apelin ELISA kit (Abclonal, Woburn, MA, USA) according to the manufacturers’ protocols. Absorbance was measured at 450 nm (iMark Microplate reader, Bio-Rad, Hercules, CA, USA). The concentrations obtained were recalculated per cell count and were used in the analysis as concentrations per 1 × 10^6^ cells. Statistical analysis of the data was performed using an unpaired two-tailed *t*-test or a two-way ANOVA, with multiple comparisons corrected with a Bonferroni test (GraphPad Prism 8.0.1. software). The results were presented as mean ± SEM. The differences were considered to be statistically significant at * *p* < 0.05, ** *p* < 0.01, *** *p* < 0.001, **** *p* < 0.0001. The results on apelin secreted by neurons were calculated using the formula: Sn—Sg, where Sn is the final apelin concentration in culture medium, and Sg is the amount of the protein in glial mixed medium that was used for neuron cultivation.

### 3.6. Quantitative Real-Time PCR (qPCR)

Total RNA was extracted from the cells with a TriReagent (MRC, Cincinnati, OH, USA) following the manufacturer’s instructions, with subsequent DNA-free DNA Removal Kit (Invitrogen, Carlsbad, CA, USA) treatment. cDNA was synthesized on 0.5–2 μg of total RNA using M-MLV Reverse Transcriptase (Evrogen, Moscow, RF, Russia) with random primers. The primer sequences are shown in Table 7. The cDNA obtained was amplified using LightCycler 96 instrument (Roche, Basel, Switzerland) set to the following reaction conditions: denaturation (preincubation) at 95 °C (5 min), amplification cycles n = 45 (94 °C, 20 s; 55–65 °C, 20 s; 72 °C, 20 s), melting (95 °C, 10 s; 65 °C, 60 s; 97 °C, 1 s), cooling at 37 °C, 30 s. qPCRmix-HS SYBR reaction mixture (Evrogen, Moscow, RF, Russia) was used. 18S rRNA was accepted as a reference. Changes in expression levels of target genes were calculated using the 2^−∆∆Ct^ method. A statistical analysis of the qPCR data was performed using an unpaired two-tailed *t*-test and aa two-way ANOVA, with multiple comparisons corrected with a Bonferroni test (GraphPad Prism 8.0.1. software). The differences were considered statistically significant at * *p* < 0.05, ** *p* < 0.01, *** *p* < 0.001, **** *p* < 0.0001.

## 4. Conclusions

Glia-derived soluble factors influenced inflammatory signaling pathways in neurons at transcriptional and post-transcriptional levels, and those changes were dependent on both components of the system, i.e., on neurons and glia traits, which indicates close humoral interactions between these two cell types.

If one or both of the cell types is characterized by Parkin deficiency, the interaction resulted in the general suppression of the inflammatory gene network in neurons, which might represent a defense reaction. Stress-induced up-regulation of *APLNR* was a characteristic feature of PD neurons and was diminished by HD and PD soluble glial factors. When both glia and neurons lacked Parkin, which referred to natural conditions in PD, *APLNR* transcription was depressed; however, in this case, neurons increased *APLN* expression and also apelin synthesis and apelin release into intracellular fluid, demonstrating one more compensatory action.

Overall, the reported results indicate that neurons have self-defense mechanisms contributing to cell survival in Parkin deficiency. To our knowledge, for the first time, the level of apelin secretion by HD neurons and PD neurons with impaired Parkin function under the influence of HD and PD glial soluble factors has been determined.

Our results refer to the conditions when glia is not activated and produce small amounts of inflammatory cytokines/chemokines. As PD is characterized by increased production of inflammatory molecules by activated microglia and astrocytes, further comparative research on neuronal responses to inflammatory stimuli at low and high concentrations is needed.

The findings presented not only contribute to the understanding of PD mechanisms in all their complexity, but also support the prospects of research into therapeutic methods targeting the apelin/APLNR system and NF-kB signaling in neurons, since they reproduce natural defense mechanisms in Parkin-deficient neurons.

## Figures and Tables

**Figure 1 ijms-25-09621-f001:**
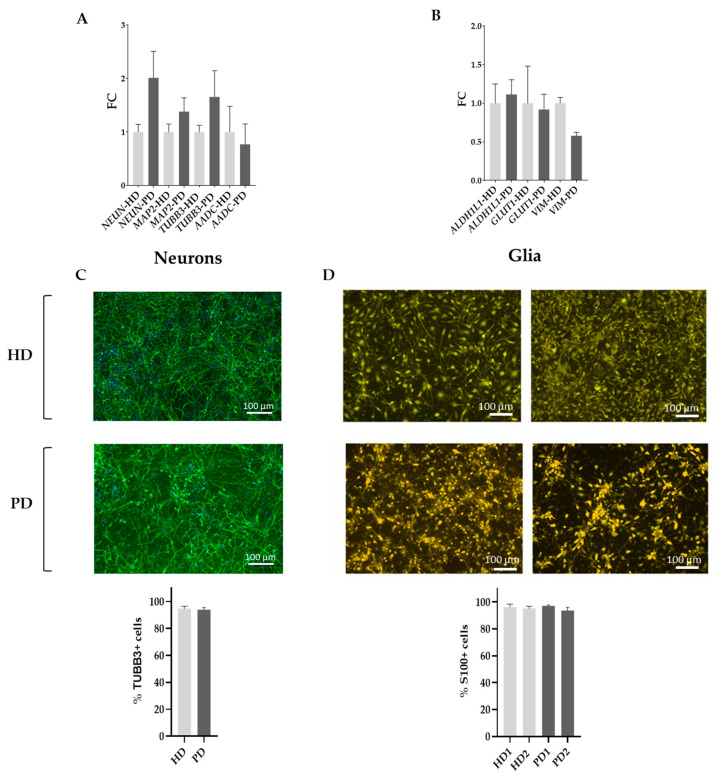
Differentiation status of neuronal and glial cell cultures used in the study. (**A**,**B**), qPCR data on neuronal markers in iPSC-derived neuronal cultures and astrocytic markers in iPSC-derived glial cultures normalized to the mean level of gene expression in HD. Light-gray columns—HD cell lines, dark-gray columns—PD cell lines. Statistical analysis was performed using unpaired two-tailed *t*-test (GraphPad Prism 8.0.1. software). The differences were considered statistically non-significant at *p* > 0.05. (**C**,**D**), Immunocytochemical analysis of TUBB3 expression in neuronal cell lines and S100 expression in glial cell lines. Green—TUBB3, yellow—S100, blue—DAPI. Graphs show % of TUBB3- and S100-positive cells in the cell cultures. Light-gray columns—HD cell lines, dark-gray columns—PD cell lines. The differences were considered statistically non-significant at *p* > 0.05.

**Figure 2 ijms-25-09621-f002:**
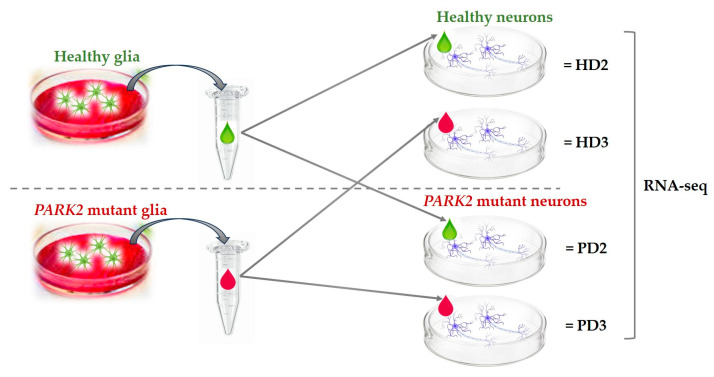
The experimental design. HD neurons and PD neurons with *PARK2* mutations were cultivated in mixed conditioned medium obtained from two HD glial cultures (“2”) and in mixed conditioned medium obtained from two PD glial cultures with *PARK2* mutations (“3”). HD2—HD neurons in HD glial mixed medium, PD2— PD neurons in HD glial mixed medium, HD3— HD neurons in PD glial mixed medium, PD3— PD neurons in PD glial mixed medium.

**Figure 3 ijms-25-09621-f003:**
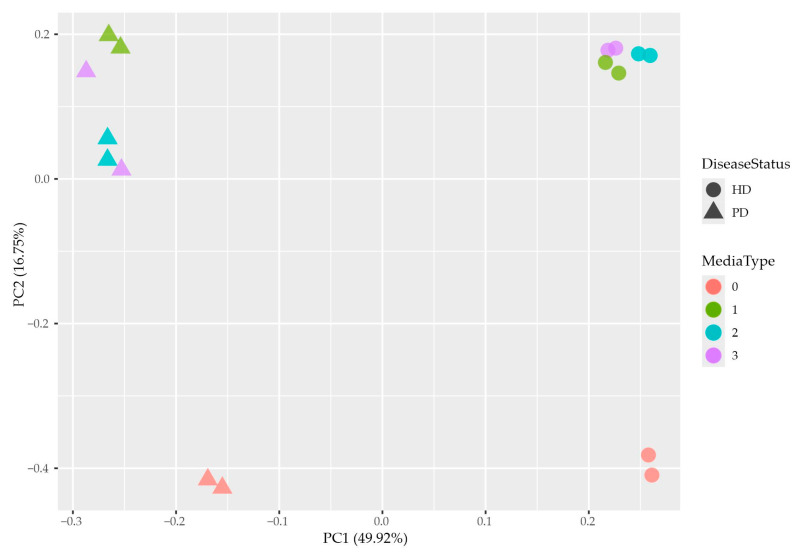
Principal Component Analysis of RNA-seq samples. HD—neurons of healthy donors, PD—neurons of PD patients; 0–3—the type of culture medium: 0—base neuronal medium, 1—base glial medium, 2—HD glial mixed medium, 3— PD glial mixed medium.

**Figure 4 ijms-25-09621-f004:**
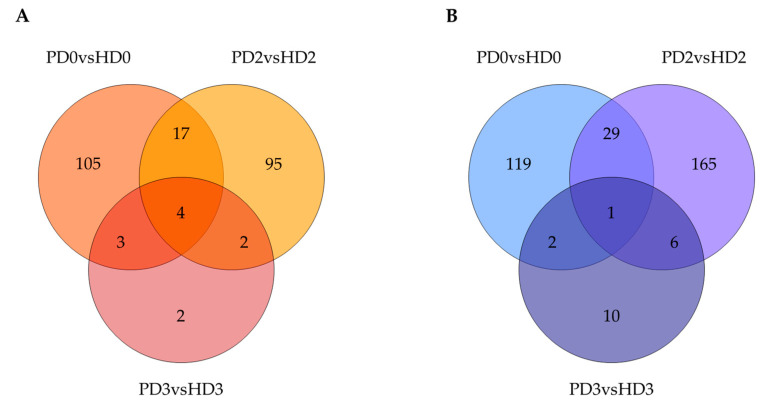
Venn diagram showing the overlay of DEGs (FC = 1.5) in different comparisons. (**A**), up-regulated genes; (**B**), down-regulated genes. The numbers correspond to the number of DEGs.

**Figure 5 ijms-25-09621-f005:**
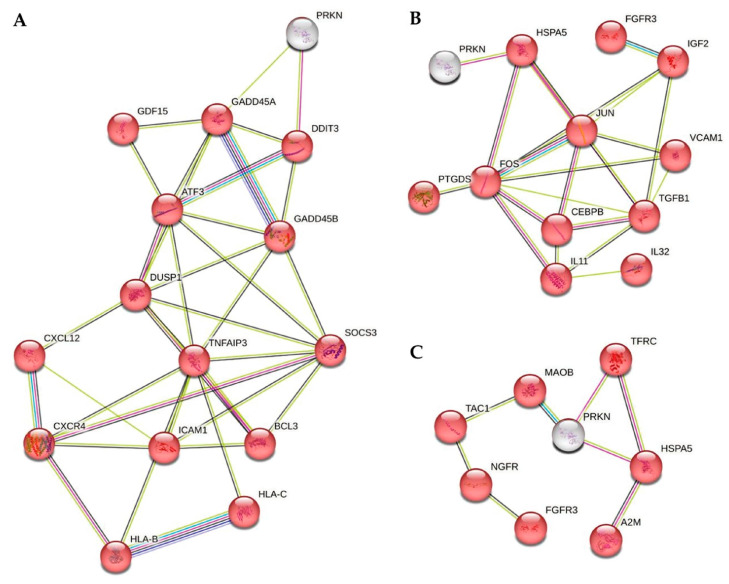
Protein network between the products of key DEGs and Parkin (PRKN) in different Groups. (**A**) Group 1; (**B**) Group 2; (**C**) Group 3. STRING (Search Tool for the Retrieval of Inter-acting Genes/Proteins). Line color indicates the type of interaction evidence: light blue—from curated databases, dark blue – gene co-occurrence, pink—experimentally determined, green—from text mining, gray – co-expression, violet – protein homology.

**Figure 6 ijms-25-09621-f006:**
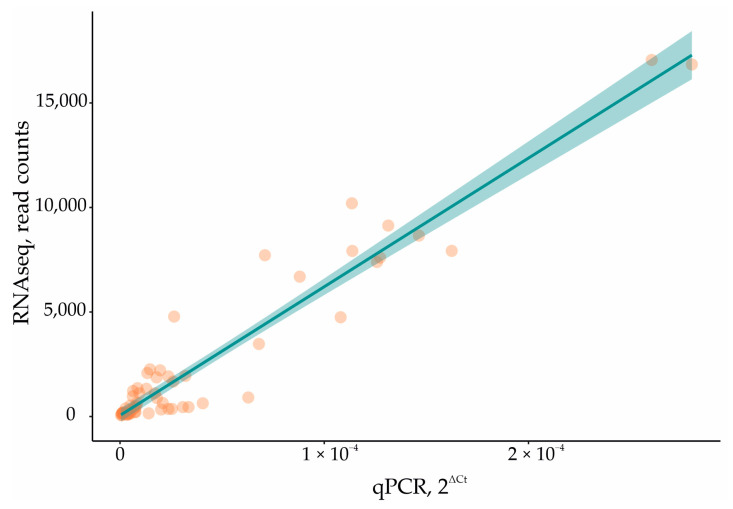
Relationship between the RNAseq and qPCR data. Regression model statistics: F-statistic 704.1 on 1 and 62 DF; *p*-value < 2.2 × 10^−16^; adj. R^2^ = 0.92.

**Figure 7 ijms-25-09621-f007:**
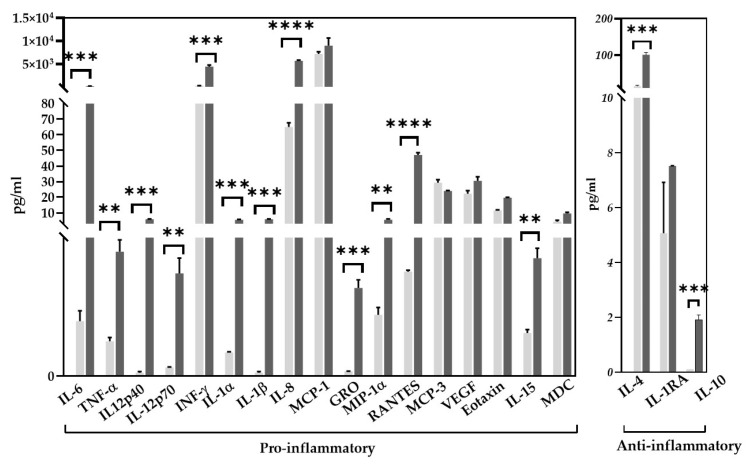
Cytokine/chemokine composition of HD and PD glial mixed supernatants. Multiplex analysis. Light-gray columns—concentrations in HD glial mixed supernatant, dark-gray columns—concentrations in PD glial mixed supernatant. The results are presented as mean ± SEM (samples were taken in three technical replicates). Statistical analysis was performed using unpaired two-tailed *t*-test (GraphPad Prism 8.0.1. software). The differences were considered statistically significant at ** *p* < 0.01, *** *p* < 0.001, **** *p* < 0.0001.

**Figure 8 ijms-25-09621-f008:**
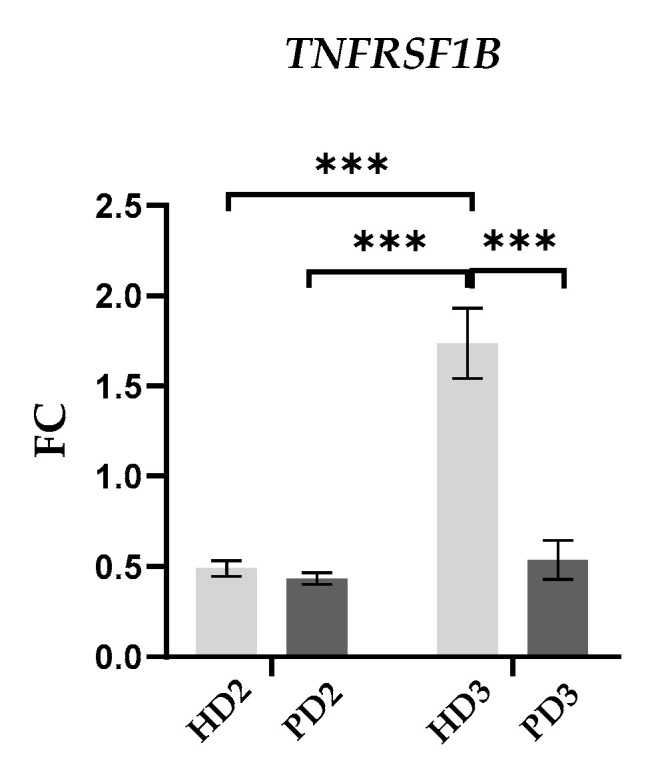
The expression of *TNFRSF1B* in HD and PD neurons cultivated in HD and PD glial mixed media. qPCR results were normalized to the mean level of the gene expression in HD1. Light-gray—HD neurons, dark-gray—PD neurons; 2—HD glial mixed medium, 3—PD glial mixed medium. Statistical analysis of the data was performed using two-way ANOVA, with multiple comparisons corrected with Bonferroni test (GraphPad Prism 8.0.1. software). The differences were considered statistically significant at *** *p* < 0.001.

**Figure 9 ijms-25-09621-f009:**
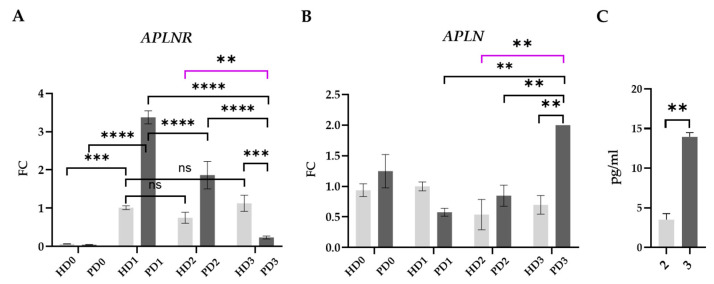
Apelin/APLNR signaling components. (**A**,**B**), qPCR data on the expression of *APLNR* and *APLN* in neurons normalized to the mean level of gene expression in HD1. Light-gray—HD neurons, dark-gray—PD neurons; 0—base neuronal medium, 1—base glial medium, 2—HD glial mixed medium, 3—PD glial mixed medium. The results are presented as mean ± SEM. Statistical analysis was performed using two-way ANOVA, with multiple comparisons corrected with Bonferroni test. The differences were considered statistically significant at ** *p* < 0.01, *** *p* < 0.001, **** *p* < 0.0001, ns—non-significant differences. Purple line corresponds to the differences between HD2 and PD3, i.e., between neurons in the “own system”. (**C**), Apelin levels in HD and PD glial mixed media. ELISA (three technical replicates). Light-gray—HD neurons, dark-gray—PD neurons; 0—base neuronal medium, 1—base glial medium, 2—HD glial mixed medium, 3—PD glial mixed medium. The results are presented as mean ± SEM. Statistical analysis was performed using unpaired two-tailed *t*-test. The differences were considered statistically significant at ** *p* < 0.01.

**Figure 10 ijms-25-09621-f010:**
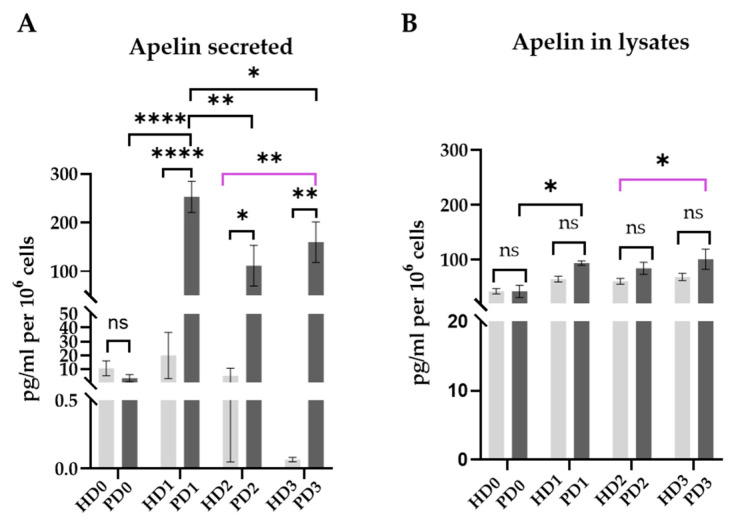
Apelin protein levels. ELISA. (**A**), Apelin secreted by neurons into intracellular fluid; (**B**), apelin in cell lysates (see Section 3). Light-gray—HD neurons, dark-gray—PD neurons; 0—neuronal medium, 1—base glial medium, 2—HD glial mixed medium, 3—PD glial mixed medium. The results are presented as mean ± SEM. Statistical analysis of the data was performed using two-way ANOVA, with multiple comparisons corrected with Bonferroni test (GraphPad Prism 8.0.1. software). The differences were considered statistically significant at * *p* < 0.05, ** *p* < 0.01, **** *p* < 0.0001, ns—non-significant differences. Purple line corresponds to the differences between HD2 and PD3, i.e. between neurons in the “own system”.

**Table 1 ijms-25-09621-t001:** Cell lines description.

Disease Status	Donor	Differentiation Status	Genotype	iPSC Cell Line Symbol
HD	Healthy female, 18 years	Neurons	normal	IPSHD1.1S [20]
PD	PD female, disease onset 30 years, biopsy 41 years	Neurons	*hom EX8 del PARK2*	IPSPDS13 [21]
PD	PD male, disease onset 38 years, biopsy 40 years	Neurons	*het EX2 del PARK2*	IPSPDS14 [22]
HD	Healthy female, 26 years	Astrocytes	normal	IPSFD3.9L [20]
HD	Healthy female, 18 years	Astrocytes	normal	IPSHD1.1S [20]
PD	PD male, disease onset 40 years, biopsy 54 years	Astrocytes	*het EX2 dup PARK2*	IPSPDS12 [23]
PD	PD male, disease onset 38 years, biopsy 40 years	Astrocytes	*het EX2 del PARK2*	IPSPDS14 [22]

**Table 2 ijms-25-09621-t002:** Numbers of DEGs in different comparisons.

Comparison	HD Neurons			PD Neurons		
	Up	Down	All	Up	Down	All
HD glial mixed medium vs. base glial medium	2057	2045	4102	510	632	1142
PD glial mixed medium vs. base glial medium	273	133	406	581	392	973
PD glial mixed medium vs. HD glial mixed medium	2111	1983	4094	521	296	817

Logarithmic fold change (LFC) = 0; *p* < 0.05; down—down-regulated, up—up-regulated, all—total number of DEGs.

**Table 3 ijms-25-09621-t003:** Description of comparisons that were used in the analysis of RNAseq data.

Short Comparison Name	Scheme of Bioinformatic Data Analysis	Comparison Meaning
PD0vsHD0	(PD0vsHD0) vs. (PD1vsHD1)	The differences between PD and HD neurons in the base neuronal medium.
PD2vsHD2	(PD2vsHD2) vs. (PD1vsHD1)	The differences in the influence of secretome of HD glia on PD and HD neurons.
PD3vsHD3	(PD3vsHD3) vs. (PD1vsHD1)	The differences in the influence of secretome of PD glia on PD and HD neurons.
HD3vsHD2	(HD3vsHD1) vs. (HD2vsHD1)	The differences in the influence of secretome of HD and PD glia on HD neurons.
PD3vsPD2	(PD3vsPD1) vs. (PD2vsPD1)	The differences in the influence of secretome of HD and PD glia on PD neurons.
PD3vsHD2	(PD3vsHD2) vs. (PD1vsHD1)	The differences between neurons in the “own system”: PD neurons in PD glial mixed medium and HD neurons in HD glial mixed medium.

HD—neurons of healthy donors, PD—neurons of PD patients; 0–3—the type of culture medium: 0—base neuronal medium, 1—base glial medium, 2—HD glial mixed medium, 3—PD glial mixed medium. The influence of the base glial medium was taken into account in all of the comparisons.

**Table 4 ijms-25-09621-t004:** Inflammation-related pathways enriched in different comparisons. GSEA and ORA results.

mjnb	Pathway	PD0vsHD0	PD3vsHD2	PD2vsHD2	HD3vsHD2	PD3vsHD3	PD3vsPD2
GO:0034135	regulation of Toll-like receptor 2 signaling pathway			enriched			
GO:0034140	negative regulation of Toll-like receptor 3 signaling pathway		enriched	enriched	enriched		
GO:0038061	non-canonical NF-kappaB signal transduction			enriched			
GO:0000165	MAPK cascade		enriched	enriched			
GO:0038066	p38MAPK cascade		enriched	enriched	enriched		
GO:0051896	regulation of phosphatidylinositol 3-kinase/protein kinase B signal transduction				enriched		
GO:0035976	transcription factor AP-1 complex			enriched			
GO:1990622	CHOP-ATF3 complex		enriched	enriched			
GO:0036488	CHOP-C/EBP complex			enriched			
GO:0038160	CXCL12-activated CXCR4 signaling pathway		enriched	enriched			
GO:0070371	ERK1 and ERK2 cascade			enriched			
GO:0006959	humoral immune response			enriched	enriched		
GO:0019955	cytokine binding			enriched			
GO:0004896	cytokine receptor activity			enriched			
GO:0005125	cytokine activity		enriched	enriched	enriched		
GO:0019221	cytokine-mediated signaling pathway		enriched	enriched			
GO:1990868	response to chemokine				enriched		
GO:0034097	response to cytokine		enriched	enriched	enriched		
GO:0070555	response to interleukin-1			enriched			
GO:0034612	response to tumor necrosis factor			enriched			
GO:0032611	interleukin-1 beta production			enriched			
GO:0032613	interleukin-10 production			enriched			
GO:0032623	interleukin-2 production			enriched			
GO:0032633	interleukin-4 production			enriched			

Red color—pathways enriched in the comparison.

**Table 5 ijms-25-09621-t005:** Key DEGs association with neuroinflammation and PD, based on the literature data.

Gene	Annotation	Influence on Neuroinflammation	Association with Neurodegeneration, PD and/or Parkin
DEGs associated with disease status of neurons			
*CXCL12*/*CXCR4*	C-X-C Motif Chemokine Ligand 12	Cause MAPK1/MAPK3 activation and release of pro-inflammatory cytokines [25].	Up-regulated in post-mortem brains of PD individuals [26].
*DDIT3*	DNA Damage Inducible Transcript 3	Promotes NLRP3 inflammasome formation [27].	NRLP3 is a substrate for ubiquitination by Parkin [28]. * Down-regulated in SH-SY5Y cells upon expression of the WT and the mutant Parkin compared to cells not expressing Parkin [29]. *
*ICAM1*	Intracellular Adhesion Molecule 1	Is a ligand for leucocyte adhesion protein [30].	Suppressing ICAM-1 leads to a significant decrease in inflammation in MPTP PD mouse model [31].
*HLA-B* *HLA-C*	Members of Major Histocompatibility Complex, Class I	Monitors the state of the organism by the immune system, presentation of mitochondrial antigens [32].	DA neurons express MHC class I molecules in presence of activated microglia or other stressors linked to inflammation, becoming “visible” to the immune system; in the absence of Parkin, high levels of mitochondrial antigens are presented on MHC class I molecules in both macrophages and dendritic cells [32]. *
*TNFAIP3*	TNF Alpha Induced Protein 3	Terminates the NF-κB acting as ubiquitin-editing enzyme [33].	Down-regulated in whole blood from PD patients [34]. Reduced in the SN of PD patients and mice and in the MPP+-treated SK-N-SH cells; plays a neuroprotective role in MPTP-induced mice by restricting NF-κB and mTOR pathways [33].
*SOCS3*	Suppressor of Cytokine Signaling 3	Regulates a STAT3-mediated chemokine and chemokine receptor function, causes degradation of p65 NF-κB [35,36].	In PD, SOCS signaling is inhibited, which leads to neuroinflammation and subsequent degeneration of DA neurons [37].
*DUSP1*	Dual Specificity Phosphatase 1	Inactivates MAPKs by dephosphorylation [38].	Up-regulated in PD [39]. LRRK2 was found to interact with DUSP1 and DUSP16 [40].
*ATF3*	Activating Transcription Factor 3	Suppresses the expression of various inflammatory genes induced by TLR4 signaling, including genes encoding IL-6, IL-12b, and TNF-α [41].	Up-regulated in midbrain neurons treated with 6-OHDA; neurons from ATF3 “knockout” mice were not protected from cell death [42].
*BCL3*	BCL3 Transcription Coactivator	Regulates the activity of NF-κB depending on the stimulation [43].	Presented in the Hereditary Parkinsonism Gene Set [44].
*GDF15*	Growth Differentiation Factor 15	Acts as an inflammatory protein, but also have protective and trophic functions for neurons [45].	Promotes survival of DA neurons and regulated the inflammatory response after intrastriatal 6-OHDA lesion [46].
*GADD45A*	Growth Arrest and DNA Damage Inducible Alpha	Is neuroprotective and can prevent neurodegeneration [47].	The lack of Gadd45a gene expression exacerbates neuroinflammation [47]. Up-regulated early after intrastriatal 6-OHDA lesion in rats specifically in substantia nigra DA neurons [48]. Parkin overexpression reduces *GADD45A* expression in ceramide-treated PC12 cells [49]. *
DEGs characteristic for PD neurons in HD glial mixed medium			
*JUNB/FOS*	AP-1 Transcription Factor Subunits	Activate IL-6 and TNF-α production [50].	AP-1 promotes the vicious cycle of neuroinflammation in microglia of the SNpc, driving DA neurons toward apoptosis [37].
*CEBPB*	CCAAT Enhancer Binding Protein Beta	Acts as a positive regulator of cytokine gene expression [41].	Depletion of C/EBPβ diminishes DA neuronal loss in human α-SNCA transgenic mice. C/EBPβ/AEP pathway correlates with Lewy body pathologies in human PD patients [51].
*VCAM1*	Vascular Cell Adhesion Molecule 1	Promotes leukocyte adhesion and transmigration across the vascular endothelium; IL-1β and TNFα up-regulated VCAM-1 [52].	Levels of the soluble VLA4 ligand VCAM1 are highly increased in patients with PD [53]. In cultured human lung endothelial cells, down-regulation of Parkin by siRNA decreases LPS-induced VCAM-1 expression [54]. *
*IGF2*	Insulin Like Growth Factor 2	Reduces microgliosis, decreases the expression of pro-inflammatory cytokines [55].	IGF2 plasma levels and IGF2 mRNA and protein levels are significantly decreased in PBMCs in PD patients; a relationship between IGF2 levels and the autophagy process in PD was found [56].
*TGFB1*	Transforming Growth Factor Beta 1	Represses inflammatory response [57].	Neuroprotective TGF-β signaling pathway is down-regulated during development of PD [37]. TGF-β1 expression is increased in biopsies of PD patients [58].
*FGFR3*	Fibroblast Growth Factor Receptor 3	Activates MAPK and IL6 and IL8 production [59].	Neuronal expression of FGFR3 is increased in post-mortem Lewy body disease brains [60].
*HSPA5*	Heat Shock Protein Family A (Hsp70) Member 5	Performs quality control of proteins, suppresses α-synuclein aggregation and toxicity [61].	GRP78/BIP/HSPA5 is potential target for the regulation of ER stress in experimental PD models [61].
*PTGDS*	Prostaglandin D2 Synthase	Provides anti-inflammatory and antioxidant functions [62].	Increased in twins with PD relative to healthy twins [63]. Is linked with neurodegenerative disease and brain injury; mediates anti-inflammatory effects of DJ-1, a PD-related gene [62].
DEGs characteristic for HD neurons in PD glial mixed medium			
*NGFR* *(p75NTR)*	Nerve Growth Factor Receptor	Activates NF-kB and MAPK [64].	NGFR signaling pathway induces a double-sword effect, either detrimental or beneficial depending on the ligands and status of PD neuropathology [65].
*FGFR3*	Fibroblast Growth Factor Receptor 3	Activates MAPK and IL6 and IL8 production [59].	Neuronal expression of FGFR3 was increased in post-mortem Lewy body disease brains [60].
*TAC1*	Tachykinin Precursor 1	Is responsible for the high density of microglia, up-regulates inflammation [66].	SP encoded by *TAC1* gene exacerbated DA cell death; in the 6-OHDA-induced model of PD, the administration of additional SP accelerated disease progression [66].
*TFRC*	Transferrin Receptor	Is the primary iron import mechanism; induces α-synuclein fibril formation by ROS production [67].	Strongly up-regulated in TPFPD-treated (a chronic-type inflammatory stimulation combining PD patient-derived αSYN fibrils, TNFα and prostaglandin E2) human induced-microglial-like cells [67].
*MAOB*	Monoamine Oxidase B	Is associated with reactive astrogliosis [68].	Comprehensively reviewed MAOB’s role as a DA-degrading enzyme and as a critical enzyme regulating astrocytic reactivity [68]. Stable transfection of Parkin in the human DA neuroblastoma cell line SH-SY5Y markedly reduced the activities of MAOB [69]. *
*A2M*	Alpha-2-Macroglobulin	Binds pro-inflammatory cytokines, cytokine clearance [70].	Increased plasma A2M level in AD patients; A2M participated in delay of PD neuroinflammation [71].
*HSPA5*	Heat Shock Protein Family A (Hsp70) Member 5	Performs quality control of proteins, suppresses α-synuclein aggregation and toxicity [61].	GRP78/BIP/HSPA5 is potential target for the regulation of ER stress in experimental PD models [61].

* Related to Parkin.

**Table 6 ijms-25-09621-t006:** Antibodies used in the study.

Marker	Antibodies	Dilution	Company, Cat #
Primary antibodies	anti-TUBB3	1:2000	Abcam, #AB7751
	Rabbit anti-S100	1:2	Agilent Dako, #GA50461-2
Secondary antibodies	Goat anti-Rabbit IgG (H + L), AF546	1:1000	TermoFisher, #A11010
	Goat anti-Mouse IgG (H + L), AF488	1:1000	TermoFisher, #A11008

**Table 7 ijms-25-09621-t007:** The sequences of primers used in the study.

Gene	Forward	Reverse	Annealing t °C
*TNFAIP3*	CTGAAAACGAACGGTGACGG	CCATGGGTGTGTCTGTGGAA	60
*DDIT3*	CACCTCCTGGAAATGAAGAGGAAG	CTCTGGGAGGTCTTGTGAC	60
*GADD45A*	GAATTCTCGGCTGGAGAGCA	ACGTTATCGGGGTCGACGTT	65
*ATF3*	CTCTGCGCTGGAATCAGTCA	CTTCTTCAGGGGCTACCTCG	60
*NGFR*	TGCCAGGACAAGCAGAACAC	CCAGGGATCTCCTCGCAC	60
*FGFR3*	CTGGGAGATCTTCACGCTGG	GTCCAGGTACTCGTCGGTG	60
*A2M*	AGCAGGAAGACATGAAGGGC	AATCACGTCCCCGGTAGGTA	60
*HSPA5*	CATCAACGAGCCTACGGCA	AGACACATCGAAGGTTCCGC	65
*APLNR*	GTCAGCCAGCTACTCTTCGG	AGG GGATGGATTTCTCGTGC	60
*APLN*	GCTGGAAGACGGCAATGTCC	GCGGAATTTCCTCCGACCT	60
*TUBB3*	CTCAGGGGCCTTTGGACATC	CAGGCAGTCGCAGTTTTCAC	60
*NEUN*	TACGCAGCCTACAGATACGCTC	TGGTTCCAATGCTGTAGGTCGC	60
*AADC*	CTCGGACCAAAGTGATCCAT	GGGTGGCAACCATAAAGAAA	60
*MAP2*	AGGCTGTAGCAGTCCTGAAAGG	CTTCCTCCACTGTGACAGTCTG	60
*GLUT1*	GCTGTGCTTATGGGCTTCTC	CACATACATGGGCACAAAGC	60
*VIM*	ATTCCACTTTGCGTTCAAGG	CTTCAGAGAGAGGAAGCCGA	60
*TNRSF1B*	CACCGGGAGCTCAGATTCTT	CCGAAAGGCACATTCCTCCT	60
*18S*	CGGCTACCACATCCAAGGAA	GCTGGAATTACCGCGGCT	60

## Data Availability

The data discussed in this publication have been deposited in NCBI’s Gene Expression Omnibus [102] and are accessible through GEO Series accession number GSE272394 (https://www.ncbi.nlm.nih.gov/geo/query/acc.cgi?acc=GSE272394 accessed on 21 July 2024).

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
