# Peer review of "Inflammatory Intracellular Signaling in Neurons Is Influenced by Glial Soluble Factors in iPSC-Based Cell Model of PARK2-Associated Parkinson’s Disease"

_ijms, 2024, doi:10.3390/ijms25179621_

Round 1

Reviewer 1 Report

Comments and Suggestions for Authors

Parkinson's disease (PD) is a progressive neurodegenerative disorder characterized by motor symptoms such as tremors, rigidity, bradykinesia, and postural instability. Neuroinflammation is a prominent feature of Parkinson’s disease (PD), contributing significantly to the disease's progression and pathogenesis. Chronic inflammation in the brain is characterized by the activation of glial cells, particularly microglia and astrocytes, and the release of various pro-inflammatory cytokines. This manuscript used neuronal and glial cell cultures differentiated from iPSC of healthy donors (HD) and PD patients to evaluate the influence of glial secretome on inflammatory gene networks. These data increased our knowledge about the defense mechanisms related to inflammatory intracellular pathways in astrocytes and neurons. However, there are several major and minor weaknesses in the rationale and research methods of this work. Below please find the review comments. 

(1) Major comments

1. The neuronal and glial cell cultures system can be useful to clarify the interaction of neurons and specific conditioned media, however, the RNAseq and QPCR data from these in vitro artificial system is not sufficient to measure and analyze gene expression changes within a REAL PD patient (or model). It is essential to validate the interaction results by other independent in vivo models such as animal or human samples. Moreover, the DEGs within the different comparation are far away from clinic, it is better to verify with other public PD database to make sure the results are repeatable.

2, It was unclear about the cutoff and P-value for DEG analysis in Table2, as the number of DEGs is closely correlated with these values, the up and down regulated should also be highlighted in this table and figure 4. I believe the volcano plots are better to interpret the DEGs than table 2.

(2) Minor comments

1. Please revise the abstract of the manuscript. It is important to list the background, method, results and conclusion shortly and briefly to make the reader get the main points.

2. Abbreviations should be properly defined and interpreted when they are first introduced in the manuscript. This practice ensures that readers can understand the meaning of abbreviations without confusion. It is also recommended to include a comprehensive list of abbreviations after the conclusion section. This list will serve as a quick reference for readers, enabling them to easily access the definitions throughout the manuscript.

(3) Recommendations

I do not recommend the publication of the current manuscript without adequately addressing the major and minor weaknesses identified in this review.   

Comments on the Quality of English Language

NA

Author Response

(1) Major comments

Comments 1: The neuronal and glial cell cultures system can be useful to clarify the interaction of neurons and specific conditioned media, however, the RNAseq and QPCR data from this in vitro artificial system is not sufficient to measure and analyze gene expression changes within a REAL PD patient (or model). It is essential to validate the interaction results by other independent in vivo models such as animal or human samples. Moreover, the DEGs within the different comparation are far away from clinic, it is better to verify with other public PD database to make sure the results are repeatable.

Response 1: We appreciate your careful reading of the manuscript and valuable comments.

Investigation of PD is accompanied by certain difficulties due to the lack of model systems fully representing the complex mechanisms involved in disease development. The mouse models do not represent the pathophysiological neurodegeneration and protein aggregation pattern observed in PD patients, and are thus limited. The task of extracting live neurons from patients is near impossible because of both ethical and technical issues. Patient-specific iPSCs can be differentiated into disease-relevant cell types carrying the genetic background of the donor and enabling de novo generation of human models of genetically complex disorders. Currently, this model is widely used to study the molecular mechanisms of various neurodegenerative diseases, including PD. The use of iPSCs provides an opportunity to answer important questions such as the functional significance of molecular results, the contribution of individual genetic variations, and the individual patient's response to specific interventions. iPSC-based platform has great potential for in vitro modelling of such neurodegenerative diseases as PD [1; 2; 3].

It is also important to note that our study is designed according to the goal to investigate one and specific component of PD pathogenesis – humoral influence of glial cells to neurons. Such a study seems not possible to be repeated in vivo, since it requires the presence of only two types of cells. In our opinion, such a study will be of help to better understand the complexity of PD pathogenesis.

We have added the information to the manuscript (lines 90-98; 571-574).

Ref:

  1. Valadez-Barba V, Juárez-Navarro K, Padilla-Camberos E, Díaz NF, Guerra-Mora JR, Díaz-Martínez NE. Parkinson's disease: An update on preclinical studies of induced pluripotent stem cells. Neurologia (Engl Ed). 2021 Mar 11:S0213-4853(21)00020-7. English, Spanish. doi: 10.1016/j.nrl.2021.01.005;
  2. D’Sa, K., Evans, J.R., Virdi, G.S. et al. Prediction of mechanistic subtypes of Parkinson’s using patient-derived stem cell models. Nat Mach Intell 5, 933–946 (2023). https://doi.org/10.1038/s42256-023-00702-9;
  3. Yarkova, E.S.; Grigor’eva, E.V.; Medvedev, S.P.; Pavlova, S.V.; Zakian, S.M.; Malakhova, A.A. IPSC-Derived Astrocytes Contribute to In Vitro Modeling of Parkinson’s Disease Caused by the GBA1 N370S Mutation. Int. J. Mol. Sci. 2024, 25, 327. https://doi.org/10.3390/ijms25010327.

Comments 2: It was unclear about the cutoff and P-value for DEG analysis in Table2, as the number of DEGs is closely correlated with these values, the up and down regulated should also be highlighted in this table and figure 4. I believe the volcano plots are better to interpret the DEGs than table 2.

Response 2: Thank you for the important note about the missing information. We have indicated the cutoff and p-value for DEGs analysis in Table 2 and highlighted up and down regulated DEGs in Table 2 and Figure 4.

(2) Minor comments

Comments 1: Please revise the abstract of the manuscript. It is important to list the background, method, results and conclusion shortly and briefly to make the reader get the main points.

Response 1: We revised the abstract, added the information about methods and necessary abbreviations, and tried to keep clear all the main points (lines 18-32).

Comments 2. Abbreviations should be properly defined and interpreted when they are first introduced in the manuscript. This practice ensures that readers can understand the meaning of abbreviations without confusion. It is also recommended to include a comprehensive list of abbreviations after the conclusion section. This list will serve as a quick reference for readers, enabling them to easily access the definitions throughout the manuscript.

Response 2: Thank you for the valuable suggestion. We put the list of abbreviations after the Conclusion section (lines 576-582). We also added a list of alphanumeric notations for samples to the description of the experimental design (lines 151-162) to make the text easier to understand when reading further.

Reviewer 2 Report

Comments and Suggestions for Authors

The authors study the inflammatory intracellular signaling in neurons affected by soluble glial factors in conditioned medium in iPSC-derived PD cell model with PARK2 gene mutations. They find that glial cell-derived soluble factors can affect inflammatory signaling pathways in neurons at transcriptional and post-transcriptional levels. These findings are interesting. However, some concerns are raised:

1, for differentiated neurons and glial cells, the authors have not validated with specific protein markers. So it can not learn how much percentage of cells have been differentiated to neurons and glial cells. whether is there any other cell type contamination in their experiment. the authors should perform the validation and verification study of their generated neurons and glial cells,

2, in Fig 9, the levels of TNFRSF1B should be doubled confirmed at protein levels, either by western blot and Elisa assay.

3, what is the significance of current findings, which can add to PD pathogenesis or therapy? The authors need to highlight it.

Author Response

Comments 1: for differentiated neurons and glial cells, the authors have not validated with specific protein markers. So, it cannot learn how much percentage of cells have been differentiated to neurons and glial cells. whether is there any other cell type contamination in their experiment. the authors should perform the validation and verification study of their generated neurons and glial cells,

Response 1: iPSCs underwent the full course of differentiation in neuronal or glial direction according to the protocols that were successfully used and published by our team in previous works [Novosadova, E.V., 2016; Novosadova, E.V., 2020]. Resulting cells expressed markers of mature neurons (NEUN, MAP2, TUBB3 and AADC) or mature astrocytes’ markers (ALDH1L1, GLUT1 and VIM) with no significant difference between HD and PD cells, which was confirmed by qPCR results (Figure 1A, B). The expression of specific protein marker of differentiated DA-neurons TUBB3 was estimated using immunocytochemistry, the marker was displayed by 93-95% of cells (Figure 1C). Glial cultures were stained with antibodies to astrocyte specific marker S100 and were shown to contain 94-97% S100-positive cells, which meant they were astrocytes (Figure 1D). We added graphs demonstrated ICH data to Figure 1.

Comments 2: in Fig 9, the levels of TNFRSF1B should be doubled confirmed at protein levels, either by western blot and Elisa assay.

Response 2: We agree with this comment. However, based on RNAseq data, the level of TNFRSF1B expression in our neuronal cultures is very low, which corresponds to the data presented in public resources (https://www.proteinatlas.org/ENSG00000028137-TNFRSF1B/tissue). That is why we don’t expect adequate results in ELISA tests. Immunocytochemistry is not informative enough for quantitative comparisons, especially when minor differences are involved.

Overall, in our study we have made a conclusion: “The most genes encoding receptors to the studied cytokines were expressed on the same level in PD and HD neurons and were not affected by soluble factors of conditioned glial media. Thus, no signs of feed-back regulation were detected.” and presented the data on TNFRSF1B as an exception, not considering it a major finding.

Comments 3: what is the significance of current findings, which can add to PD pathogenesis or therapy? The authors need to highlight it.

Response 3: Thank you for this important advice. We have added the information to the Conclusions section of the manuscript.

Reviewer 3 Report

Comments and Suggestions for Authors

Review on the manuscript of Gerasimova T et al.: “Inflammatory intracellular signaling in neurons is influenced by glial soluble factors in iPSC-based cell model of PARK2-associated Parkinson’s disease”.

In this study, the Authors explored the influence of the glial secretome on the neuronal inflammatory gene network and the impact of Parkin on this interaction. The Authors show that neurons from healthy donors (HD) or Parkinson’s disease patients with different PARK2 mutations (PD) cultivated in HD and PD glial conditioned medium exhibit transcriptional alterations, as revealed by RNA sequencing analysis. Moreover, the lack of Parkin in neurons or astrocytes resulted in a neuronal down-regulation of genes involved in inflammatory pathways. Additionally, Parkin-deficient neurons showed elevated levels of apelin, which is considered protective in various physiological and pathological contexts. Overall, this indicates that Parkin deficiency has a neuroprotective effect.

Overall, I find this topic to be of great interest, as the interplay between Parkin dysfunction and neuroinflammation is a critical factor in the pathogenesis of PD. Therefore, understanding the mechanisms linking Parkin, mitochondrial dysfunction, and neuroinflammation is crucial for developing targeted therapies for PD. I believe the Authors have addressed the main question proposed. The issues I have identified with the current form of the manuscript are listed below. I hope the Authors find the following comments and suggestions useful.

1 - For the data shown in Figure 1, graphs A and B, I recommend that the Authors represent the data as 2-(∆∆Ct), which is the standard representation of qPCR data. This way, the results for HD will always be 1, and the PD data will be relative to HD, avoiding the fluctuation seen in the graphs. This is also valid for the data shown in Figure 6B, 9B, 10C-D.

2 - Regarding the information: “In addition, expression of TUBB3 protein was estimated using immunocytochemistry, the marker was displayed by 93-95% of cells (Figure 1C)”, please include a graph in the figure showing the percentage of HD and PD neurons expressing TUBB3.

3 - Regarding the information: “Glial cultures were also stained with antibodies to astrocyte specific marker S100 and were shown to contain 95.5 ± 0.7% S100-positive cells, which meant they were mature astrocytes (Figure 1D)”, please include a graph in the figure showing the percentage of HD and PD astrocytes expressing the astrocyte specific marker S100.

4 - Regarding the information “Further, we cultivated neurons in conditioned media obtained from glial cultures in four possible combinations: HD neurons in the conditioned medium from HD astrocytes (HD2), PD neurons in the conditioned medium from HD astrocytes (PD2), HD neurons in the conditioned medium from PD astrocytes (HD3), and PD neurons in the conditioned medium from PD astrocytes (PD3) (Figure 2)”. At this point, it is not clear that “2” refers to a mix of conditioned media obtained from HD glial cultures and that “3” refers to a mix of conditioned media obtained from PD glial cultures. This information only appears for the first time in the legend of Figure 3. Therefore, I recommend that the Authors provide this clarification earlier when describing this information.

5 - It appears that a mix of conditioned media obtained from HD and PD glial cultures was used. There are 2 donors of PD neurons, 2 donors of HD astrocytes, and 2 donors of PD astrocytes. Are all of these loss-of-function mutations? Was a mix of neurons from the 2 PD donors used? Was a mix of astrocytes from the 2 HD/PD donors used? Particularly for the PD donors, they have distinct alterations in the Parkin gene. Therefore, one alteration may trigger effects distinct from the other. This information is highly relevant and should be mentioned in the manuscript.

6 - The Authors indicate that “On the contrary, comparisons PD3vsHD3 and PD3vsPD2 didn’t show any significant differences in the inflammatory gene network”. For these results, the Authors provide two explanations: “This fact might presumably be explained either by a higher saturation of HD glial supernatants with soluble factors or by a higher activity of functional receptors on plasma membranes of HD neurons”. If HD glial supernatants are saturated with soluble factors, the PD3vsPD2 comparison should show differences, since they were comparing PD neurons in the presence of conditioned medium from PD astrocytes (PD3) versus HD astrocytes (PD2). Alternatively, if there is higher activity of functional receptors on the plasma membranes of HD neurons, the PD3vsHD3 comparison should also be significantly different, since they were comparing PD and HD neurons cultured in the conditioned medium of PD astrocytes. Therefore, these hypotheses cannot explain these results.

7 - For the data shown in Figure 6, the Authors used base neuronal medium (“0”), HD glial medium (“2”), and PD glial medium (“3”). However, the correct control should be base glial medium instead of base neuronal medium, because the other comparisons used conditioned glial medium. Could the Authors clarify this point?

8 - Regarding the data shown in Figure 8 and Figure S3, the differences between them are not clear. The data represented in Figure 8 came from 3 replicates, while the data shown in Figure S3 came from 2 biological replicates. Could the Authors clarify what they mean by “replicates” and “biological replicates”? Why are there two graphs showing the same results? Couldn't the results be combined into a single graph?

9 - In general, the manuscript is difficult to understand and follow. Therefore, I suggest that the Authors simplify some ideas and present them in a clearer way. Combining the results and the discussion does not make it easier to follow the findings.

Comments on the Quality of English Language

 Minor editing of English language required

Author Response

Comments 1: For the data shown in Figure 1, graphs A and B, I recommend that the Authors represent the data as 2-(∆∆Ct), which is the standard representation of qPCR data. This way, the results for HD will always be 1, and the PD data will be relative to HD, avoiding the fluctuation seen in the graphs. This is also valid for the data shown in Figure 6B, 9B, 10C-D.

Response 1: Thank you for your kind advice. We applied this method of representation to all the Figures containing qPCR results. The data on neuronal and glial differentiation markers was normalized to the mean level of gene expression in HD, and the data on DEGs expression – to the mean level of gene expression in HD1.  We did not use normalization PD to the level of HD in corresponding medium (to get HD results to be 1), since the difference between medium types would be lost in this case. We also presented RNAseq and PCR data together on the same graphs in Figure 6 (it is Figure S4 in revised manuscript). We hope that in this form it is more clear that qPCR results confirm RNAseq data.

Comments 2: Regarding the information: “In addition, expression of TUBB3 protein was estimated using immunocytochemistry, the marker was displayed by 93-95% of cells (Figure 1C)”, please include a graph in the figure showing the percentage of HD and PD neurons expressing TUBB3.

Response 2: We agree with the comment. We included a graph showing the expression of TUBB3 protein (ICH results) results in the Figure 1.

Comments 3: Regarding the information: “Glial cultures were also stained with antibodies to astrocyte specific marker S100 and were shown to contain 95.5 ± 0.7% S100-positive cells, which meant they were mature astrocytes (Figure 1D)”, please include a graph in the figure showing the percentage of HD and PD astrocytes expressing the astrocyte specific marker S100.

Response 3: We included a graph showing the expression of S100 protein by glial cultures (ICH results) in the Figure 1.

Comments 4: Regarding the information “Further, we cultivated neurons in conditioned media obtained from glial cultures in four possible combinations: HD neurons in the conditioned medium from HD astrocytes (HD2), PD neurons in the conditioned medium from HD astrocytes (PD2), HD neurons in the conditioned medium from PD astrocytes (HD3), and PD neurons in the conditioned medium from PD astrocytes (PD3) (Figure 2)”. At this point, it is not clear that “2” refers to a mix of conditioned media obtained from HD glial cultures and that “3” refers to a mix of conditioned media obtained from PD glial cultures. This information only appears for the first time in the legend of Figure 3. Therefore, I recommend that the Authors provide this clarification earlier when describing this information.

Response 4: Thank you for pointing this out. We added a list of alphanumeric notations for samples to the description of the experimental design (lines 151-162) to make further reading easier.

Comments 5:

It appears that a mix of conditioned media obtained from HD and PD glial cultures was used. There are 2 donors of PD neurons, 2 donors of HD astrocytes, and 2 donors of PD astrocytes. Are all of these loss-of-function mutations?

Response 5: We assumed that the function of Parkin was impaired in the cells with all the mutations investigated, since the patients were diagnosed with the Parkinson's disease.

Was a mix of neurons from the 2 PD donors used?

No, neuronal cultures from different PD patients were cultivated and investigated separately, i.e. in different experiments (lines 351-354, Figures S4, S5).

Was a mix of astrocytes from the 2 HD/PD donors used?

No, we cultivated glial cultures from PD each donor separately, collected conditioned medium from each cell culture, and then prepared mixed HD supernatant (1:1) and mixed PD supernatant (1:1). Finally, two types of glial mixed medium (HD and PD) for neurons cultivation were prepared, they contained 70% of mentioned above glial mixed supernatants + 30% of base neuronal medium. We revised correspondent section in Materials and Methods carefully (lines 588-606).

Particularly for the PD donors, they have distinct alterations in the Parkin gene. Therefore, one alteration may trigger effects distinct from the other. This information is highly relevant and should be mentioned in the manuscript.

We agree that effects of mutations might be different. However, the function of Parkin was impaired in all PD cell cultures as all the mutations caused PD in patients. That's why we assumed that the processes occur in a similar way. Independent experiments with two different PD cultures demonstrated corresponding results, which was confirmed by qPCR (Figure S4 and Figure S5).

Comments 6: The Authors indicate that “On the contrary, comparisons PD3vsHD3 and PD3vsPD2 didn’t show any significant differences in the inflammatory gene network”. For these results, the Authors provide two explanations: “This fact might presumably be explained either by a higher saturation of HD glial supernatants with soluble factors or by a higher activity of functional receptors on plasma membranes of HD neurons”. If HD glial supernatants are saturated with soluble factors, the PD3vsPD2 comparison should show differences, since they were comparing PD neurons in the presence of conditioned medium from PD astrocytes (PD3) versus HD astrocytes (PD2). Alternatively, if there is higher activity of functional receptors on the plasma membranes of HD neurons, the PD3vsHD3 comparison should also be significantly different, since they were comparing PD and HD neurons cultured in the conditioned medium of PD astrocytes. Therefore, these hypotheses cannot explain these results.

Response 6: Thank you for the important note. We deleted this hypothesis from the text. The only conclusion can be made is that in the field of inflammation HD cell system was more reactive than PD one, as neurons and glia obtained from iPSCs of HDs had greater ability to interact with each other (lines 239-241).

Comments 7: For the data shown in Figure 6, the Authors used base neuronal medium (“0”), HD glial medium (“2”), and PD glial medium (“3”). However, the correct control should be base glial medium instead of base neuronal medium, because the other comparisons used conditioned glial medium. Could the Authors clarify this point?

Response 7: Figure 6 (it is Figure S4 in revised manuscript) demonstrates that qPCR results correspond to RNAseq data. Actually, there is no control here. Your question gave us an idea that it might be better to indicate samples in all the four types of media and to put RNAseq and qPCR data together on the same graph. We also normalized the data in this Figure (and qPCR data in all other figures) on the mean level of gene expression in HD1.

Comments 8: Regarding the data shown in Figure 8 and Figure S3, the differences between them are not clear. The data represented in Figure 8 came from 3 replicates, while the data shown in Figure S3 came from 2 biological replicates. Could the Authors clarify what they mean by “replicates” and “biological replicates”? Why are there two graphs showing the same results? Couldn't the results be combined into a single graph?

Response 8: The data in Figure 8 (it is Figure 7 in revised manuscript) refers to cytokine/chemokine concentrations in HD and PD mixed conditioned media, specifically the ones we prepared and used for neurons cultivation. Three technical replicates were taken for analysis from each of the two mixed media. 

In order to make sure that not only PD mix but both initial PD glial supernatants contained higher levels of cytokines than HD ones, we conducted the additional round of multiplex protein analysis with supernatants from individual glial cell cultures (2 HD and 2 PD) taken in 2 biological replicates. The results correlated to concentration ratio between HD and PD mix media (Figure S3).

This information is presented in lines 358-361, 375-379 and in the captions of Figure 7 and Figure S3.

Comments 9: In general, the manuscript is difficult to understand and follow. Therefore, I suggest that the Authors simplify some ideas and present them in a clearer way. Combining the results and the discussion does not make it easier to follow the findings.

Response 9: Thank you for the advice. The study is quite complex and bioinformatic algorithm is not simple. In our opinion, combining the results and the discussion sections allows readers to trace the sequence of research, as the results of one stage influence the next one. We tried to do our best and made additional thorough revision of the text, and we hope, that made the manuscript easier to follow.

Round 2

Reviewer 1 Report

Comments and Suggestions for Authors

I appreciate the author's responses to my queries and the additional interpretation they have done to address these points. It is regrettable that no insights were added to understand the mechanism of PD. I feel the revised manuscript still needs improve before publication. 

Author Response

Comments: I appreciate the author's responses to my queries and the additional interpretation they have done to address these points. It is regrettable that no insights were added to understand the mechanism of PD. I feel the revised manuscript still needs improve before publication.

Response: We have carefully re-examined our findings and reformulated the conclusions so that they more fully reflected the contribution of our study to the general understanding of PD mechanisms (lines 23-31; 495-532; 549-551; 565-576).

At the same time, it was important for us to avoid speculations and not to draw unfounded conclusions. We have correlated our data with publications on key DEGs activity and association with PD and/or Parkin (Table 5), as well as with those on the features of different signaling pathways. However, we would not like to make a direct comparison of our data with the results of published studies conducted in animal models and with the data of public transcriptomic databases of patients with PD.  The design of the research is artificial, since PD neurons in vivo are never surrounded by HD glia, but such a model system has many advantages: neurons and glia are differentiated from patient-specific iPSCs; PARK2-mutations, differentiation status and Parkin loss of function are confirmed in the cell cultures; the system contains just two cell types which allows to determine the presence of mutual influence; the model allows to use an appropriate control for bioinformatic analysis. It is also important to take into account the conditions in which the research is carried out in order to make conclusions and to point further direction for research (577-581).

We sincerely believe that our study has brought the new information to the general understanding of PD pathogenesis regarding self-protective mechanisms in neurons, such as suppression of intracellular inflammatory pathways (including endogenous NF-kB signaling), activation of APLNR/APLN expression and apelin synthesis. To the best of our knowledge, these observations are novel (lines 574-576). The results also support the studies using apelin and NF-kB inhibitors as therapeutic tools for PD correction (lines 582-585).

Reviewer 2 Report

Comments and Suggestions for Authors

The revised manuscript has been significantly improved, so can be accepted for publication now.

Author Response

We are grateful to the Reviewer for the valuable help in improving the manuscript.

Reviewer 3 Report

Comments and Suggestions for Authors

Second review on the manuscript of Gerasimova T et al.: “Inflammatory intracellular signaling in neurons is influenced by glial soluble factors in iPSC-based cell model of PARK2-associated Parkinson’s disease”.

In this study, the Authors explored the influence of the glial secretome on the neuronal inflammatory gene network and the impact of Parkin on this interaction. The Authors show that neurons from healthy donors (HD) or Parkinson’s disease patients with different PARK2 mutations (PD) cultivated in HD and PD glial conditioned medium exhibit transcriptional alterations, as revealed by RNA sequencing analysis. Moreover, the lack of Parkin in neurons or astrocytes resulted in a neuronal down-regulation of genes involved in inflammatory pathways. Additionally, Parkin-deficient neurons showed elevated levels of apelin, which is considered protective in various physiological and pathological contexts. Overall, this indicates that Parkin deficiency has a neuroprotective effect.

The represents the second version of the manuscript after peer review. After a careful reding of the revised manuscript, I consider that some issues still need clarification, which are listed below. I hope the Authors find the following comments and suggestions useful.

1 - During the first revision process, the Authors were recommended to represent the data shown in Figure 1, graphs A and B, as 2(-∆∆Ct) (also called fold change). The Authors responded by stating that this method was applied to all figures containing qPCR results and that the data on neuronal and glial differentiation markers were normalized to the mean level of gene expression in HD. However, upon examining the graphs, the data appear to be represented as 2ct and not as 2(-∆∆Ct). Additionally, if the data on neuronal and glial differentiation markers were normalized to the mean level of gene expression in HD, the results for gene expression in HD should be 1, which is not the case. I recommend that the Authors clarify this point. This is also valid for all the data of qPCR

2 - For the data shown in Figure S4 (old Figure 6), the Authors used base neuronal medium (“0”), HD glial medium (“2”), and PD glial medium (“3”). Why base neuronal medium was used instead of base glial medium? This way, the Authors are comparing 1 condition in neuronal medium with 2 conditions in glial medium, which is not correct. Could the Authors clarify this point?

Comments on the Quality of English Language

Minor editing of English language required.

Author Response

Comments 1a: During the first revision process, the Authors were recommended to represent the data shown in Figure 1, graphs A and B, as 2(-∆∆Ct) (also called fold change). The Authors responded by stating that this method was applied to all figures containing qPCR results and that the data on neuronal and glial differentiation markers were normalized to the mean level of gene expression in HD. However, upon examining the graphs, the data appear to be represented as 2ct and not as 2(-∆∆Ct).

Response 1a: Sorry for a technical mistake. We accidentally embedded a non-final version of the Figure 1 into the manuscript. Thank you for pointing it out.

Comments 1b: If the data on neuronal and glial differentiation markers were normalized to the mean level of gene expression in HD, the results for gene expression in HD should be 1, which is not the case. I recommend that the Authors clarify this point. This is also valid for all the data of qPCR

Response 1b: That is correct, the data on neuronal and glial differentiation markers were normalized to the mean level of gene expression in HD (line 133) . We analyzed the data in several replicates (about 6), and the replicates differ from each other, so that the resulting columns for gene expression in HD were not equal 1 but fluctuated around 1. Such approach allowed us to assess the statistical non-significance (p>0.05) of the differences in differentiation markers between HD and PD. This also applies for all the qPCR data that was normalized to the mean level of gene expression in HD1.  

Comments 2: For the data shown in Figure S4 (old Figure 6), the Authors used base neuronal medium (“0”), HD glial medium (“2”), and PD glial medium (“3”). Why base neuronal medium was used instead of base glial medium? This way, the Authors are comparing 1 condition in neuronal medium with 2 conditions in glial medium, which is not correct. Could the Authors clarify this point?

Response 2: All the PCR data in Figure S4 (old Figure 6) was normalized to the mean level of gene expression in HD1 that corresponds to neurons in base glial medium (lines 725-726). We did not use normalization of the expression in PD to the level of HD in each medium type separately (to get HD = 1), since the difference between medium types would be lost in this case. Figure S4 includes samples in all the four types of media and demonstrates that qPCR results correspond to RNAseq data. There is no control here.

We have also checked and corrected the English language.

We are gateful to the Reviewer for the valuable help in improving the manuscript.

Round 3

Reviewer 1 Report

Comments and Suggestions for Authors

I appreciate the author's responses to my queries and the additional experimental work they have done to address these.